# Lipidome atlas of the adult human brain

**Maria Osetrova** [1], **Anna Tkachev**[1], **Waltraud Mair**[1], **Patricia Guijarro Larraz**[1], **Olga Efimova** [1], **Ilia Kurochkin**[1], **Elena Stekolshchikova**[1], **Nickolay Anikanov**[1], **Juat Chin Foo**[2], **Amaury Cazenave-Gassiot** [2], **Aleksandra Mitina**[1], **Polina Ogurtsova**[1], **Song Guo**[1], **Daria M. Potashnikova**[3], **Alexander A. Gulin**[4], **Alexander A. Vasin**[4,5], **Anastasia Sarycheva**[1], **Gleb Vladimirov**[1], **Maria Fedorova**[6], **Yury Kostyukevich**[1], **Evgeny Nikolaev**[1], **Markus R. Wenk**[2] ✉, **Ekaterina E. Khrameeva** [1] ✉ & **Philipp Khaitovich**[1] ✉

Lipids are the most abundant but poorly explored components of the human brain. Here, we present a lipidome map of the human brain comprising 75 regions, including 52 neocortical ones. The lipidome composition varies greatly among the brain regions, affecting 93% of the 419 analyzed lipids. These differences reflect the brain's structural characteristics, such as myelin content (345 lipids) and cell type composition (353 lipids), but also functional traits: functional connectivity (76 lipids) and information processing hierarchy (60 lipids). Combining lipid composition and mRNA expression data further enhances functional connectivity association. Biochemically, lipids linked with structural and functional brain features display distinct lipid class distribution, unsaturation extent, and prevalence of omega-3 and omega-6 fatty acid residues. We verified our conclusions by parallel analysis of three adult macaque brains, targeted analysis of 216 lipids, mass spectrometry imaging, and lipidome assessment of sorted murine neurons.

The human brain indisputably represents one of the most complex biological structures. Current efforts to characterize its detailed organization tend to follow two directions: (i) molecular studies focusing on the brain region macroscale architecture mainly represented by gene expression analysis of individual cells, brain structures, and isolated cell populations[1–6]; and (ii) macroscale studies of the structural and functional organization of the brain mainly based on brain imaging techniques[7,8]. Several layers of the human brain organization potentially bridging the two research streams, however, largely escape the current scientific focus. Among these layers, we believe, is the human brain lipidome.

Lipids are the main components of brain tissue, comprising 78% of the dry weight of axon myelin sheath and 35-40% of the neuron-rich gray matter[9]. Over the past 150 years, research on the human brain lipidome has provided substantial information on the biochemical composition of selected gray and white matter regions, axon-wrapping myelin, and major brain cell types[10–14]. These pioneering studies identified a multitude of lipid classes comprising brain tissue, listing phosphatidylcholines and phosphatidylethanolamines representing phospholipids, ceramides representing sphingolipids, and cholesterols representing sterols among its main components. These studies further revealed substantial variation of lipid class content and fatty acid composition among isolated brain structures including, among others, higher levels of sphingolipids, particularly ceramides, cholesterol, and oleic acid in myelin membranes of oligodendrocyte cells compared to other brain components, neutral lipid deposits in astrocytes, and particular lipid class and fatty acid profiles of synaptic membranes. More recently, mass spectrometry-based methods have provided compound-level resolution of lipids in gray and white matter, selected brain regions, and cultured brain cells from humans[15,16],

¹Skolkovo Institute of Science and Technology, Moscow, Russia. ²Singapore Lipidomics Incubator, Life Sciences Institute and Precision Medicine Translational Research Program, Department of Biochemistry, Yong Loo Lin School of Medicine; National University of Singapore, Singapore, Singapore. ³Department of Cell Biology and Histology, Faculty of Biology, Lomonosov Moscow State University, Moscow, Russia. ⁴N. N. Semenov Federal Research Center for Chemical Physics Russian Academy of Sciences, Moscow, Russia. ⁵Department of Chemistry, Lomonosov Moscow State University, Moscow, Russia. ⁶Leipzig University, Leipzig, Germany. ✉e-mail: bchmrw@nus.edu.sg; e.khrameeva@skoltech.ru; p.khaitovich@skoltech.ru

macaques[17,18], and mice[19–21]. Building on previous work, these studies have identified additional differences in lipid composition among brain regions, cell types, and cellular projections.

As our understanding of the brain's lipidome composition improves, we are also gaining insight into the functional roles of lipids. The lipid class ratio, the size and the charge of lipid head groups and the number, length and unsaturation extent of their fatty acid residues define geometry, fluidity, and compartmentalization of membrane bilayers, including formation of structured lipid rafts[22–25]. In addition to these well-established roles, lipids are now recognized to critically contribute to brain energy metabolism[26], cell type differentiation[27,28], neuronal and glial signaling[28–30], control of inflammatory response[31] and modulation of protein complexes[32]. Furthermore, alterations in the brain lipidome have been linked to cognitive disorders: autism, schizophrenia, Alzheimer's disease, alcohol-related brain damage, and others[33–37]. Rodent studies similarly showed significant aging-related brain lipidome alterations and lipid composition differences among brain regions, brain cell types, and individual brain cell types and their projections[20,21,33–37].

Despite the increasing understanding of the brain lipidome, there are still limitations in our knowledge regarding the connection between lipidome variation and the structural and functional organization of the brain. Previous lipidome studies have focused on specific brain structures comprising less than ten regions in total, limiting our ability to associate lipidome variation with data on brain structural and functional networks. To address this limitation, we conducted both untargeted and targeted lipidome characterization of 75 anatomically and functionally distinct human brain regions. The data we acquired allowed us to examine the distribution of lipid features across the brain and integrate it with the known molecular, anatomical, and functional brain features, providing a foundation for future systematic studies of the human brain's lipidome.

## Results

### Constructing lipidome map of the human brain

To construct a lipidome map of a neurotypical adult human brain, we investigated the lipidome composition in 75 anatomically and functionally distinct regions dissected from four adult cognitively healthy humans (Fig. 1a, b; Supplementary Table 1; Supplementary Data 1). In addition to humans, we assessed the brain lipidome in 38 of the 75 brain regions of three adult macaques (Fig. 1a; Supplementary Table 1; Supplementary Data 1). Unlike humans, macaques were raised in a standardized environment, followed by rapid and controlled tissue collection, thus providing a robust reference for the human brain lipidome quality evaluation.

The 75 evaluated human brain regions comprised 11 gross anatomical structures. The 56 regions represented four groups of neocortical areas: primary sensory or motor cortices (six regions), secondary cortical areas (16 regions), associative cortical areas (21 regions), and limbic cortex (13 regions). The remaining structures included basal ganglia (five regions), thalamus (four regions), hypothalamus, two midbrain regions, four dispersed white matter regions, and three cerebellar gray and white matter regions (Fig. 1a, c; Supplementary Data 1). The 38 analyzed macaque areas covered all 11 anatomical structures (Fig. 1a, c, d; Supplementary Data 1).

For all samples, we collected lipid abundance data using two complementary methods: (i) untargeted high-resolution mass-spectrometry (HRMS) and (ii) targeted tandem mass spectrometry using multiple reaction monitoring (MRM). Untargeted analysis of the human samples yielded intensities of 419 lipids annotated using ion fragmentation comprising 21 lipid classes corresponding to LIPID MAPS subclasses[38] (exceptions described in Materials and Methods) (Fig. 1b; Supplementary Fig. 1; Supplementary Data 2). MRM measurements involved 216 lipids representing the same lipids classes, with 169 shared between the techniques (Supplementary Data 2).

Macaque brain tissue measurements covered 394 of the 419 HRMS lipids and all 216 MRM lipids (Fig. 1a, b; Supplementary Data 2). To minimize the biological variation that is inevitably present among humans and, to a lesser extent, among macaque samples, we standardized average lipid abundance levels among individuals. This was achieved by dividing the abundance of each lipid in each of the 75 human or 38 macaque regions by its per-individual mean. Consequently, our subsequent analysis is based on these normalized lipid abundance levels. This approach allows us to estimate the relative representation of each lipid among the investigated brain regions. However, it reduces our ability to compare different lipids with each other in terms of their absolute abundance in the brain tissue.

Visualization of the human brain lipidome variation based on the normalized abundance of 419 HRMS-assessed or 216 MRM-assessed lipids revealed a reproducible gradient of brain structures−from associative and limbic cortical regions to the central white matter tracts (Fig. 1e, top row). The macaque lipidome measurements yielded the same gradient (Fig. 1e, bottom row). Accordingly, normalized lipid intensity varied significantly among brain regions for 391 of 419 (93%) compounds in humans and 279 of 394 (71%)−in macaques (ANOVA, BH-adjusted $p < 0.01$). Further, profiles of normalized lipid intensity differences among brain regions correlated positively and significantly between the techniques (HRMS and MRM) and the species (humans and macaques), indicating the robustness and reproducibility of our lipidome measurements (comparison to random pair correlations, Mann−Whitney $U$ test $p < 0.0001$; Fig. 1f, g; Supplementary Fig. 2,3).

Visualization of the relative representation of 21 biochemical lipid classes sorted by their geometry and lipids sorted by the extent of unsaturation in their fatty acid residues revealed distinct differences separating myelin-rich subcortical and white matter regions, as well as more subtle lipid gradients across neocortex and subcortical structures (Fig. 2a, b). For example, consistent with previous findings[10], cholesterol levels were elevated in the subcortical white matter, while lipids containing polyunsaturated fatty acids (PUFAs), such as docosahexaenoic acid (22:6), were decreased (Fig. 2a, b). In addition to this difference, both cholesterol and PUFA-containing lipids exhibited considerable variation across neocortical regions, with lower relative abundance in the prefrontal regions and elevated levels in the motor, visual, and parietal cortices.

### Lipid profiles across brain regions fall into discrete categories

Our analysis of lipid class variation among brain regions indicated that myelin content plays a substantial role in defining regional lipidome composition. This observation aligns with previous work showing a substantial contribution of myelin to the lipidome variation among cell types and brain regions in mice[20,21]. To identify lipids abundance patterns explainable by myelin representation and search for the ones displaying myelin-independent patterns, we first estimated the relative myelin content of the brain regions using human[7,39] and macaque[11,13] structural MRI (sMRI) T1w/T2w image data (Fig. 3a, b; Supplementary Data 1), recognized to provide a reasonable approximation of the myelin content[40]. For both species, myelin content distribution correlated strongly with the first principal component of the lipidome intensity distribution, even though the lipid and sMRI data were acquired from entirely distinct sets of individuals (Pearson's $R = 0.78$ and 0.77 for humans and macaques, respectively, $p < 0.0001$; Fig. 3c, d). Consistently, the intensity profiles of individual lipids correlated positively and significantly with the myelin content for 65% of detected compounds ($N = 274$; $myelin^+$ lipids) and negatively for another 17% ($N = 71$; $myelin^-$ lipids) (ANOVA, BH-corrected $p < 0.01$; Fig. 3e, Supplementary Data 2).

Comparison of $myelin^+$ and $myelin^-$ normalized lipid intensities to published human lipidome data derived from cortical white and gray matter enriched and depleted in myelin[18] verified our classification (Fisher test, $p < 0.05$; Fig. 3f). Further, direct visualization of the lipids'

intensity distributions in the human prefrontal cortical sections using MALDI imaging similarly confirmed the sMRI-based assignment of *myelin*⁺ lipids in the white matter and *myelin*⁻ lipids—in the gray matter (Fisher test, *p* < 0.05; Fig. 3g, h; Supplementary Fig. 4). Additionally, gene expression patterns correlating with lipid intensity profiles across 35 brain regions (Pearson correlation, *R* > 0.5, *p* < 0.05; Supplementary

Fig. 5) displayed enrichment in relevant Gene Ontology (GO) terms: axons and myelin—for genes correlated with *myelin*⁺ lipids, and synaptic functions—for genes correlated with *myelin*⁻ lipids (hypergeometric test, BH-corrected *p* < 0.05; Supplementary Data 3).

Among the remaining 18% of the analyzed lipidome, 11% (*N* = 46) differed among brain regions in a reproducible manner not explained

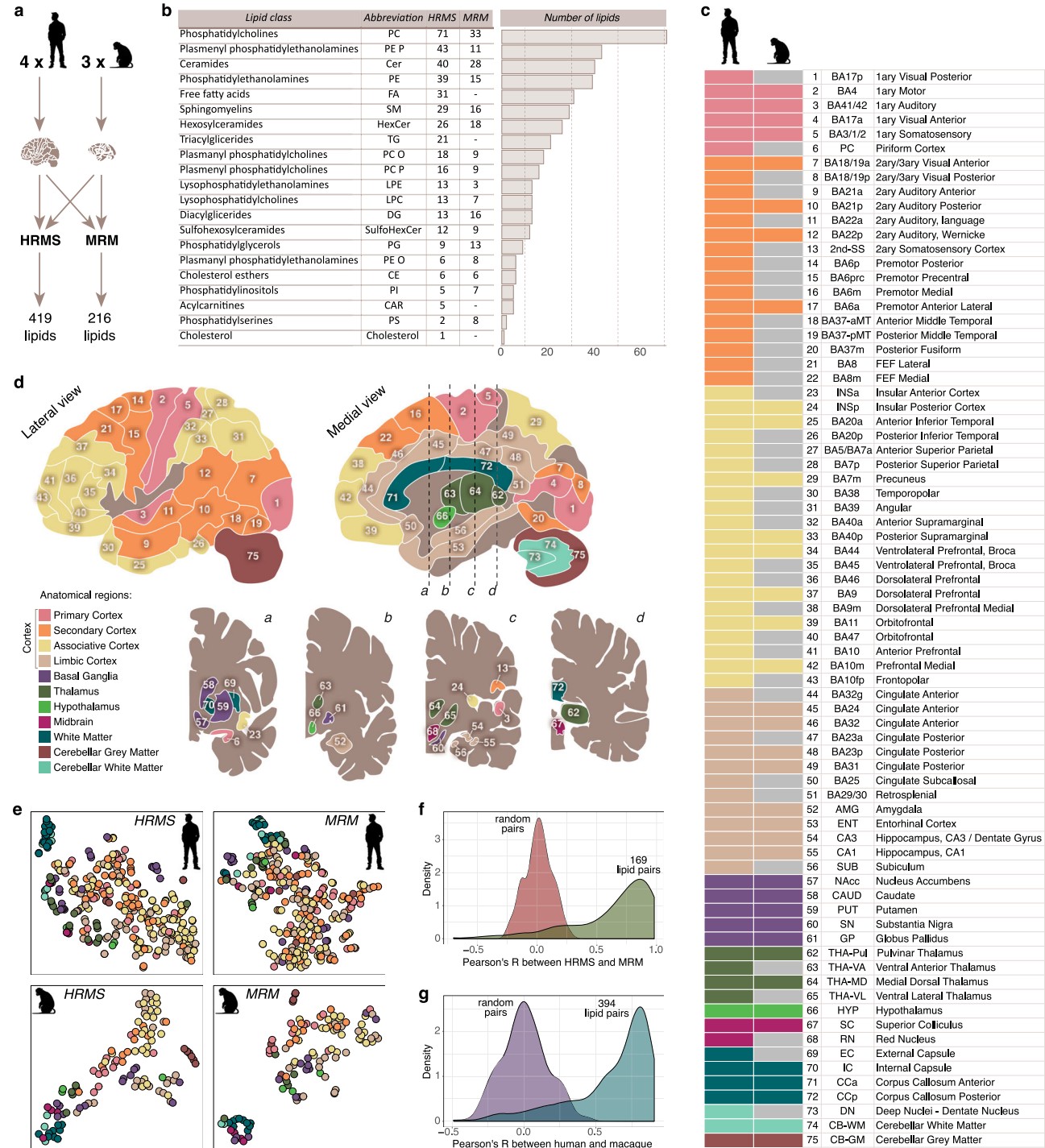

**Fig. 1 | Lipidome analysis of human and rhesus macaque brain regions.**
**a** Experimental scheme displaying numbers of subjects and lipids analyzed by mass spectrometry-based techniques (HRMS and MRM). **b** Numbers of lipids detected in the human brain by HRMS and MRM in lipid classes defined as corresponding LIPID MAPS[38] subclasses. **c, d** List of assessed brain regions (**c**) and their anatomical localization in the human brain (**d**). Brain images were adapted from ref. 1. **e** Visualization of the total lipidome variation in the human and macaque brains using t-SNE based on HRMS and MRM measurements. Each circle represents a brain region, colors according to (**d**). **f, g** Correlation of lipid intensity profiles across brain regions between HRMS and MRM (**f**) and between humans (*n* = 4 individuals) and macaques (*n* = 3 animals) (**g**). Random pairs distributions represent the correlation between lipid intensity profiles of two datasets with permuted region labels. Source data are provided as a Source Data file.

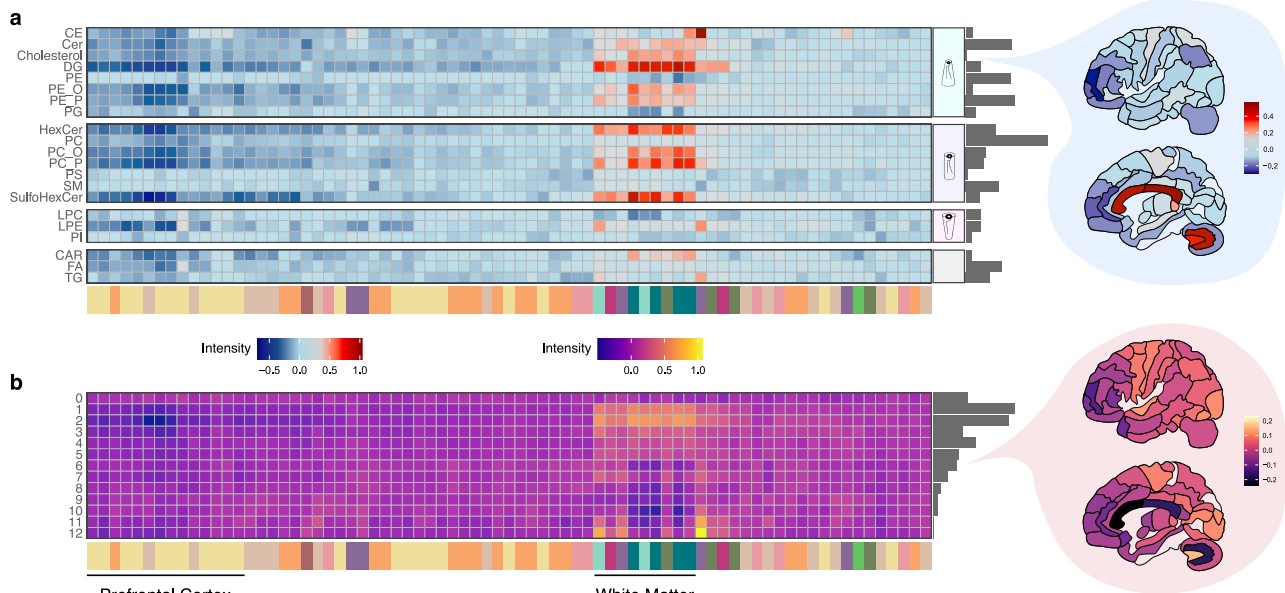

**Fig. 2 | Normalized abundance profiles of lipids sorted by class and unsaturation across brain regions. a** Heatmaps illustrating the relative abundance of each lipid class in every brain region. The normalized abundance refers to the average signal intensity of each lipid class in a given brain region normalized to its global average intensity, calculated across all 75 regions, and then averaged across individuals ($n = 4$ individuals). Each row represents a lipid class, each column corresponds to a brain region. The color bars underneath the plots here and in panel **b** indicate the anatomical assignment of brain regions as shown in Fig. 1d. Lipid classes are grouped based on their geometry, which is illustrated in an insert on the right side of the figure. The gray bars on the right side represent the number of detected lipids in each lipid class (Supplementary Data 2). The brain regions on the right side are colored according to the relative abundance levels of cholesterol. Brain images were adapted from ref. 1. **b** Heatmaps representing the normalized abundance levels of lipids containing a defined total number of double bonds in their fatty acid residues (rows) across brain regions (columns), averaged across individuals ($n = 4$ individuals). The gray bars on the right side represent the number of detected lipids in each fatty acid unsaturation group (Supplementary Data 2). The brain regions on the right side are colored according to the relative abundance levels of lipids with a total of six double bonds in their fatty acid residues. Brain images were adapted from ref. 1. Source data are provided as a Source Data file.

by the myelin content (*unexplained* lipid category). Further, 5% of the lipids ($N = 20$) did not show any intensity differences among brain regions (*housekeeping* lipids), and 2% ($N = 8$) varied substantially among individuals (*variable* lipids) (Fig. 3e; Supplementary Data 2).

Parallel analysis of the macaque brain lipidome and sMRI data yielded a consistent lipid assignment into the five pattern-based categories for 84% of the 394 overlapping HRMS-based lipids and 88% for MRM-based ones (Fig. 3i; Supplementary Fig. 6). Further, lipid intensity profiles within the categories correlated positively and significantly between the two species, with *myelin⁺* lipids showing the highest concordance, followed by *myelin⁻* and *unexplained* lipid categories (Fig. 3j; Supplementary Fig. 7).

**Biochemical properties of lipids differ among the categories**

Analysis of lipid chemical properties revealed an overrepresentation of five lipid classes among *myelin⁺* lipids, including three previously assigned to the central nervous system white[41–44] matter[41–44] (hypergeometric test, $p < 0.1$; Fig. 4a; Supplementary Data 2). The *myelin⁻* lipids showed an overrepresentation of two phospholipid classes, lysophosphatidylcholines and phosphatidylethanolamines (hypergeometric test, $p < 0.01$; Fig. 4a; Supplementary Data 2), and *housekeeping* lipids−excess of free fatty acids (hypergeometric test, $p < 0.001$; Fig. 4a; Supplementary Data 2). In addition to lipid class content, lipids comprising the pattern-based categories displayed differences in fatty acid chain unsaturation and length. Particularly, *myelin⁻* lipids tended to have polyunsaturated fatty acid residues in PE and PE-P classes (Mann−Whitney $U$ test, BH-corrected $p < 0.05$; Fig. 4a), longer residues' chain length in PE-P class and shorter−in PC class (Mann−Whitney $U$ test, BH-corrected $p < 0.1$; Fig. 4a). By contrast, *housekeeping* lipids as a category included unsaturated (Mann−Whitney $U$ test, BH-corrected $p < 0.05$; Fig. 4a) and short-chain free fatty acids (FA) (Mann−Whitney $U$ test, BH-corrected $p < 0.1$; Fig. 4a). These differences resulted in

variation in the predicted membrane fluidity, with *myelin⁻* lipids yielding the highest fluidity levels and *myelin⁺* lipids−the lowest, consistent with reported white matter properties[44] (Mann−Whitney $U$ test, BH-corrected $p < 0.001$; Fig. 4b). Experimentally, we further verified the predicted differences between categories by direct visualization of molecular ions corresponding to lipid head groups representing 12 lipid classes in human cerebellar sections using time-of-flight secondary ion mass spectrometry (ToF-SIMS), (Fig. 4b–e; Supplementary Fig. 8; Supplementary Table 2).

**Lipid profiles could be linked to particular cell types**

Analysis of primary cell cultures derived from the mouse brain reported substantial lipidome differences among major neural cell types[20,21]. To assess the contribution of cell type composition to the lipidome variation in the human brain, we first estimated the relative representation of main neural cell types (OD−oligodendrocytes, MG−microglia, OPC−oligodendrocyte progenitor cells, In−inhibitory neurons, Ex−excitatory neurons, Ast−astrocytes) using the mRNA expression of 15 established marker genes available for 35 of the 75 regions[1] (Fig. 5a). For 353 of the 419 HRMS-assessed lipids (84%), the intensity profile spanning the brain regions correlated positively and significantly with at least one marker gene expression profile (cell-type-associated lipids, Pearson's $R > 0.5$, $p < 0.01$; Supplementary Data 2). The majority (82%) of cell-type-associated *myelin⁺* lipids correlated best with the oligodendrocyte marker profiles, reflecting the fact that myelin sheaths are made by oligodendrocyte membranes (Fig. 5b, c). By contrast, the intensities of *myelin⁻* lipids correlated best with the expression profiles of inhibitory and excitatory neuron markers (Fig. 5b, c). *Housekeeping* and *variable* lipids were distributed across all cell types, while *unexplained* lipids were associated significantly with markers of astrocytes, oligodendrocyte progenitor cells, and inhibitory neurons (Fig. 5b, c) (hypergeometric test, BH-corrected $p < 0.05$).

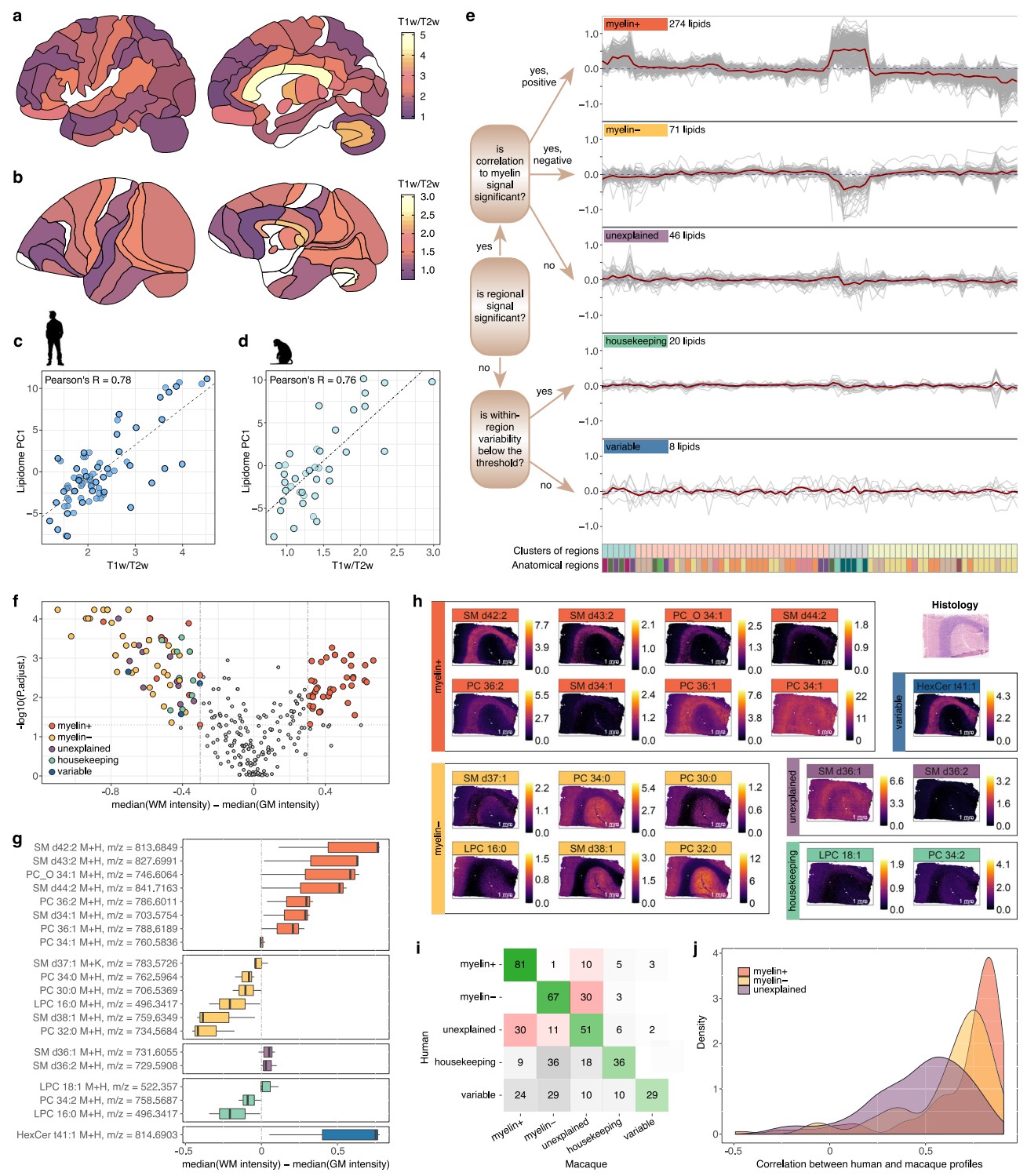

To verify the assignment of lipids to the main neural cell types, we first compared the human data to the two published mouse datasets, each covering eight of the 75 brain regions[20,21]. The comparison revealed evident and significant similarities between the lipidome variation profiles of the two species (Fig. 5d; Supplementary Fig. 9; Pearson's $R = 0.66$, $p = 0.00012$). Further, lipid classes showing significant cell-type specificity in the human brain, HexCer and PE, displayed the same cell type assignment in mice (hypergeometric test, BH-corrected $p < 0.001$; Fig. 5e, f).

We next carried out direct experimental verification of the neuronal cell type assignment by the lipidome assessment of fluorescently labeled pyramidal neurons and unlabeled cells isolated using cell

sorting from the brain of Thy1-ChR2-YFP[45] transgenic mice (Fig. 5g; Supplementary Fig. 10). The experiment confirmed the enrichment of lipids assigned to neurons in the human brain among lipids comprising sorted mouse neurons compared to sorted unlabeled cells (Wilcoxon test, $p = 0.0077$; Fig. 5h; Supplementary Fig. 10).

## Human brain lipidome features can be associated with the brain's functional architecture

Our analysis indicates that the human brain lipidome composition could be associated with the brain's structural organization features, such as myelin content and cell type composition. We next examined whether the lipidome composition of the brain could be linked to the

**Fig. 3 | Lipidome patterns within the human brain. a, b** Schematic representation of the human (**a**) and macaque (**b**) brains colored according to the relative myelin content determined using sMRI T1w/T2w image data. Brain images were adapted from[1]. **c, d** Correlation between lipidome variation based on PC1 of HRMS measurements and myelin content derived from sMRI image data in human (*n* = 210 individuals) (**c**) and macaque (*n* = 19 animals) (**d**) brains. **e** Schematic drawing of the lipid profile classification protocol and visualization of the resulting average category profiles (red line) and profiles of individual lipids within the category (gray lines). For each category plot: the *y*-axis shows normalized lipid intensity and the *x*-axis corresponds to 75 brain regions arranged according to hierarchical clustering outcome (see Fig. 2a). **f** Volcano plot showing significant lipid intensity differences between human white and gray matter calculated using public data (*n* = 5 individuals)[18] and colored according to lipid category classification from this study. *P*-values were calculated using two-sided *t* test, BH-corrected. **g, h** Visualization of spatial intensity distributions of 19 lipids detected by HRMS using MALDI imaging of human prefrontal cortical sections (*n* = 4 individuals) (**h**) and their intensities' ratio between white and gray matter areas of the sections (**g**). Here and in Figs. 4b, d, 5d, h, 6c, j, box plots represent the median, 25th and 75th percentiles; the whiskers extend to the largest and smallest value no further than 1.5 interquartile range. MALDI image panel headings indicate category placement of the lipids determined by HRMS data analysis. **i** Correspondence percentage of the lipid placement into the five categories between human (*n* = 4 individuals) and macaque (*n* = 3 animals) HRMS data. Green color indicates a perfect match, red—a mismatch, and gray—a non-contradictory alternative assignment. Color intensity reflects the number of overlapping lipids. **j** Distributions of the correlation coefficients for lipid intensity profile comparisons between humans (*n* = 4 individuals) and macaques (*n* = 3 animals) for the three main lipid categories. Source data are provided as a Source Data file.

well-described functional brain architecture features: hierarchical information processing and functional connectivity.

The hierarchy of signal processing among brain regions (HR) reflects the information flow within the brain: from the primary to the higher-level associative areas for sensory inputs and in reverse for motor commands[46]. Among the 75 brain regions, 56 could be assigned to one of four levels of the signal processing hierarchy defined according to[47] (Fig. 6a). The lipidome composition varied significantly among the four HR levels: the second principal component of the total lipid intensity variation correlated significantly with HR assignment (Pearson's *R* = 0.58, *p* = 0.0000034; Fig. 6b). The single region deviating substantially from this relationship was the piriform cortex— an entry point of the odor sensation pathway (Fig. 6b). Further, individual lipid intensities correlated with HR well above the chance expectation (HR lipids, *N* = 60, *lm(lipid intensity ~ hierarchy level)*, BH-corrected *p* < 0.05; Supplementary Data 2), 26 of them positively (HR⁺ lipids) and 34—negatively (HR⁻ lipids).

Lipids positively and negatively correlating with HR (HR⁻ and HR⁺ lipids) displayed distinct categorical, chemical, and biological properties. HR⁻ lipids were overrepresented in the *myelin⁻* category, enriched in phosphatidylcholines, and contained polyunsaturated fatty acid residues: particularly omega-3 DHA (docosahexaenoic acid) containing six double bonds (Binomial test, BH-corrected *p* < 0.05; Fig. 6c–g). By contrast, HR⁺ lipids were overrepresented in the *unexplained* category, enriched in sphingomyelins, and preferentially contained saturated or oligo-unsaturated fatty acid residues with up to four double bonds (Binomial test, BH-corrected *p* < 0.05; Fig. 6c–e; Supplementary Fig. 11). Further, in contrast to the HR⁻ group, fatty acid components of HR⁺ lipids displayed significant depletion of omega-3 and overrepresentation of omega-6 residues (Binomial test, BH-corrected *p* < 0.05; Fig. 6f). Similarly, cell type association analysis preferentially assigned HR⁺ lipids to astrocytes, OPCs, and microglia, while HR⁻ lipids – to oligodendrocytes and inhibitory neurons (Fig. 6g).

Functional brain connectivity (FC) data, reflecting the correlated activity of dispersed brain regions at rest measured using functional magnetic resonance imaging (rs-fMRI) and potentially reflecting the topology of the brain's functional networks, was available for 59 of the 75 regions[48] (Fig. 6h). The first principal component of the FC distance matrix correlated significantly with the second principal component of the normalized lipid intensities matrix, indicating the relationship between the lipidome composition and FC (Pearson's *R* = 0.35, *p* = 0.006; Fig. 6i). Accordingly, for 18% (*N* = 76) of analyzed lipids, the intensity matrices tended to correlate positively with FC ones (Mantel test, nominal *p* < 0.005). Markedly, *myelin⁺* lipids constituting the axonal tracts' sheath showed a significantly better correlation with FC than lipids in the other four categories, potentially reflecting better physical connectivity of the regions within activity-synchronized networks (Mann–Whitney *U* test, *p* = 1.5 × 10⁻⁴; Fig. 6j). Further, three of the 21 lipid classes, sulfatides, hexosylceramides, and diacylglycerols, all three enriched in *myelin⁺* lipids, demonstrated significantly higher correlation with FC than the bulk of detected lipids (Mann–Whitney *U* test, BH-corrected *p* < 0.05; Fig. 6k).

## Discussion

In this study, we aimed to bridge the gap between brain structural and functional architecture studies and microscale molecular organization analyses by constructing a systematic map of the human brain lipidome and connecting it to the brain's structural and functional organization. Previous studies have investigated the composition of the human, macaque, and rodent brain lipidome, revealing general brain lipid class and fatty acid composition, as well as identifying key biochemical features of myelin, synaptosomes, gray and white matter neocortical sections, isolated cells, and cultured brain cell types[15–21]. These studies have emphasized variations in lipid composition among different brain regions and cell types, with myelin displaying the most distinct lipidome composition. However, despite these advancements, there still exists a gap in our understanding of the specific roles played by lipids in brain structural and functional features, including cellular composition diversity and functional network architecture.

In our study, we sought to evaluate the connection between lipid abundance variation in 75 human brain regions and their structural and functional attributes. To this end, we first sorted lipids according to their structural properties using a well-established lipid class nomenclature, revealing distinct differences between myelin-rich and myelin-poor brain regions such as subcortical white matter areas and neocortical gray matter (Fig. 2). Consistent with prior studies, our analysis of lipid class and fatty acid residue unsaturation confirmed an overrepresentation of ceramides, hexosylceramides, sulfohexosylceramides, cholesterol, and lipids containing saturated fatty acid residues or ones with up to two double bonds in myelin-rich regions[41–44]. Diacylglycerides (DG) and plasmalogens of phosphatidylcholine and phosphatidylethanolamine were also found to be enriched in myelin-rich regions, corroborating previous findings[49]. Conversely, PUFA-containing lipids were concentrated in the neocortex and other myelin-poor brain regions, aligning with previous observations[50] (Fig. 2).

Given the substantial influence of myelin content on lipid abundance variations among brain regions, we developed our own lipid classification based on their distribution across the brain and the relationship between this abundance variation and myelin levels determined using structural MRI data. Our results from the human brain, confirmed by a parallel assessment using macaque brain data, indicated that the majority of detected compounds (82%) belonged to the myelin-associated profile categories, which include *myelin⁺* and *myelin⁻* lipids. Consistent with our lipid class distribution analysis, the biochemical composition of these categories aligned with published work, with sphingolipids associating positively and phospholipids containing long polyunsaturated fatty acid residues associating negatively with myelin levels[20] (Figs. 3f and 4a).

Nonetheless, despite the significant role of myelin in determining brain lipid profiles, 18% of the detected compounds were not

associated with myelin levels. Of these lipids, over a quarter did not exhibit measurable variation across the entire brain. This category, termed *housekeeping* lipids, was significantly enriched in short-chain unesterified fatty acids, aligning with their role as universal building blocks in brain lipid biosynthesis[32]. Additionally, two-thirds of the myelin-independent lipids consistently varied across both neocortical and subcortical regions, a variation that could not be explained by myelin levels and, therefore, termed as *unexplained* category.

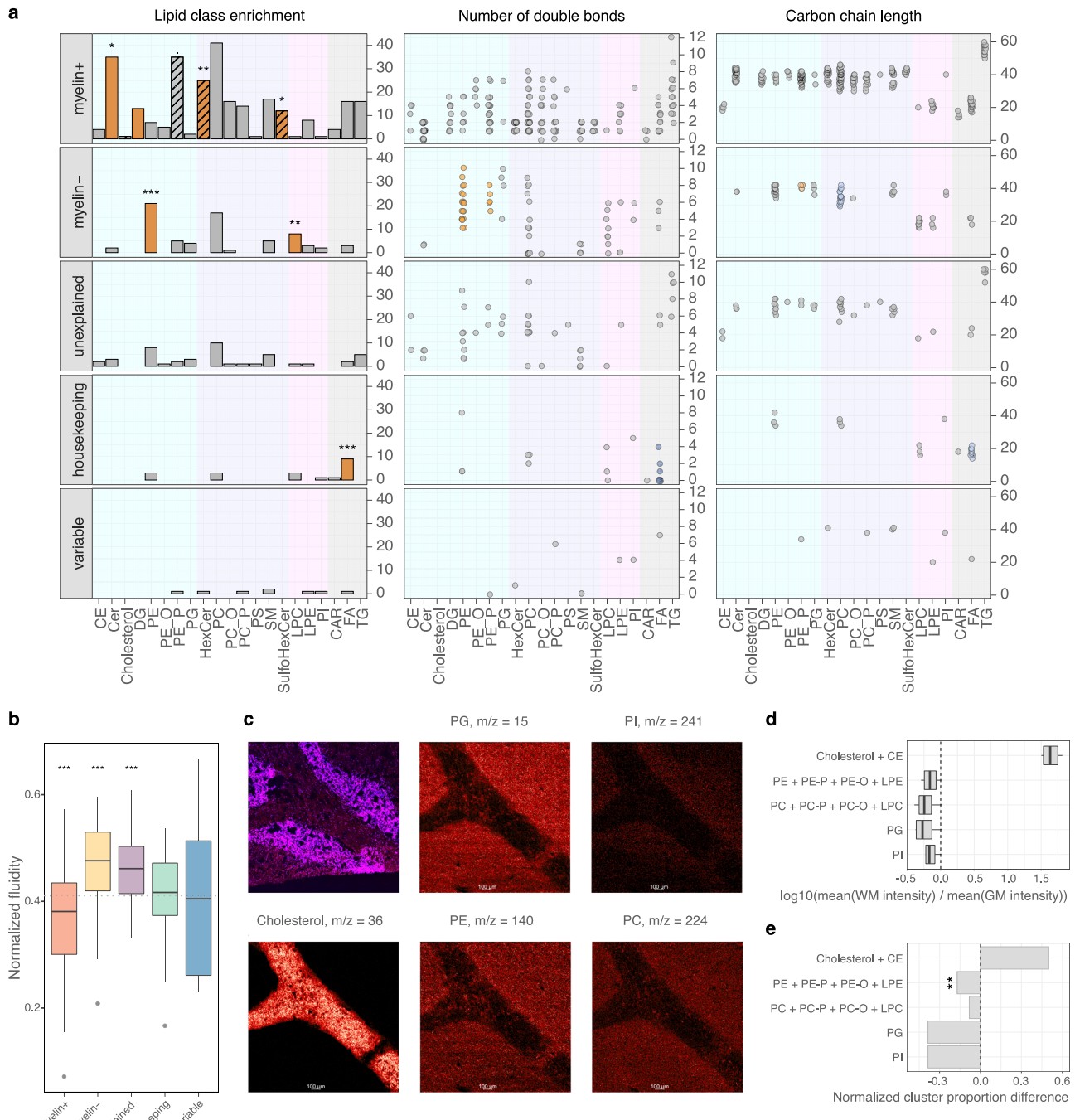

**Fig. 4 | Characterization of lipids within five brain-profile categories.**
**a** Distribution of detected lipid compounds among lipid classes within each category (left panel, total number of lipids within each category, from the top: 274, 71, 46, 20, 8) in humans (*n* = 4 individuals). Colored bars indicate significantly enriched classes (two-sided hypergeometric test, BH-corrected, ***p < 0.001; **p < 0.01; *p < 0.05; p < 0.1). Hatched bars mark lipid classes previously assigned to brain white matter. Distribution of total double bond count (central panel) and carbon chain lengths (right panel) of fatty acid residues within each lipid class in each category. Circles represent lipid compounds. Lipid compounds are represented by circles, with circle colors indicating higher (orange) or lower (blue) number of double bonds or fatty acid residues of particular length in a given class compared to the other clusters (*p < 0.05 for number of double bonds and *p < 0.1 for carbon chain length). Background color represents geometry groups as indicated in Fig. 2a. **b** Distribution of predicted membrane fluidity effects in humans (*n* = 4 individuals) for each lipid in a category (two-sided Mann–Whitney *U* test, BH-corrected, ***p < 0.001). **c, d** Visualization of the spatial intensity distributions of five lipid head group ions using SIMS imaging of human cerebellar sections (*n* = 3 individuals) (**c**) and their intensities' ratio between white and gray matter areas of the sections (**d**). **e** Difference in prevalence of lipids in *myelin+* and *myelin−* clusters shown as a proportion difference calculated for lipid class sets containing the same head group, as measured by ToF-SIMS in HRMS data (two-sided hypergeometric test, BH-corrected, **p < 0.01). Source data are provided as a Source Data file.

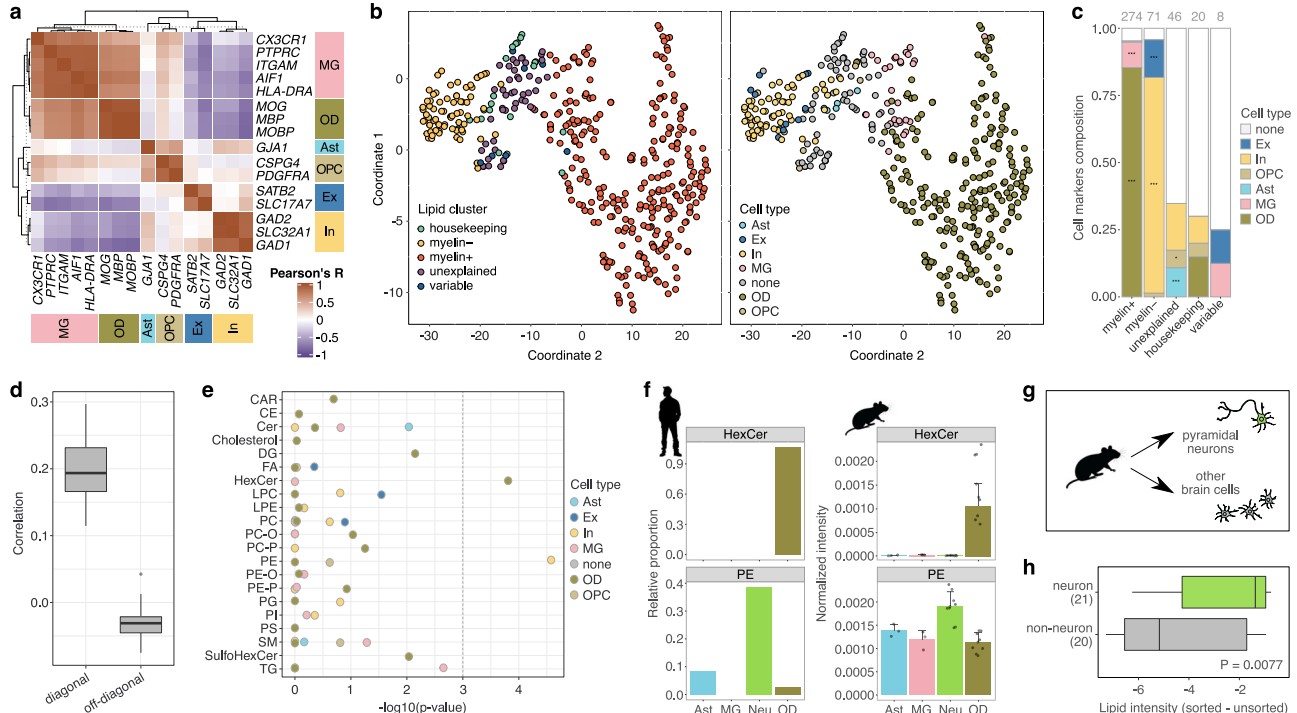

**Fig. 5 | Associations between lipid profiles and particular cell types.**
**a** Correlation of 35 brain region mRNA expression profiles of marker genes used in cell type deconvolution analysis in humans ($n = 4$ individuals here and in **b**–**e**).
**b** Clustering of 419 HRMS lipids based on their intensity profiles across brain regions using tSNE colored by category (left) and assigned cell type (right).
**c** Proportion of lipids assigned to cell types within each category and their relative enrichment (one-sided hypergeometric test, BH-corrected, ***$p < 0.0001$; **$p < 0.001$; *$p < 0.05$). **d** Distributions of correlation coefficients for human-mouse ($n = 4$ humans and $n = 3$ mice) comparisons based on lipid intensity profiles matched and mismatched between the species. **e** Lipid class association with main

brain cell types. Two lipid classes (PE and HexCer) show significant and specific associations. *P*-values were calculated using one-sided hypergeometric test, BH-corrected. **f** Comparison of the relative lipid species proportion of PE and HexCer among four cell types in the human brain ($n = 4$ individuals) and relative intensities of these two lipid classes in the corresponding cell type lines in the mouse brain ($n = 3$ animals). Error bars represent the standard deviation. **g** Scheme of mouse cell sorting experiment. **h** The intensity ratio between lipids associated with neuronal and non-neuronal cell types in our data in the lipidomes of sorted pyramidal neurons and the rest of sorted cells ($n = 2$ animals). *P*-value was calculated using one-sided Wilcoxon test. Source data are provided as a Source Data file.

To assess the functional significance of lipid level variation among brain regions, we utilized a gene expression-based tissue composition deconvolution procedure to assign lipids to the main brain cell types. This assignment process, which relied on the correlation between lipid intensity and mRNA expression profiles of cell type marker genes across 35 brain regions, proved particularly effective for myelin-dependent categories. In the other three categories, particularly among *housekeeping* and *variable* lipids that did not display consistent variation profiles across brain regions, a substantial proportion of lipids remained unassigned to specific cell types (Figs. 5c and 7). Nonetheless, our analysis suggests that the characteristic profiles of the *unexplained* lipids could be linked to the abundance of astrocytes and oligodendrocyte precursor cells in the brain regions, suggesting an association with these cell types (Fig. 5b, c and Fig. 7). The link between the *unexplained* lipid category and astrocytes was further suggested by the low conservation of their profiles between the human and macaque brains (Fig. 3i and Fig. 7). This finding aligns with the reported rapid evolution of human astrocytes, detected using single-cell transcriptome assessment[1]. By contrast, the profiles of *myelin*+ and *myelin*− lipids were highly conserved between humans and macaques and correlated with oligodendrocyte and microglia markers for the *myelin*+ and neuronal markers for the *myelin*− categories, respectively (Fig. 3f, g, Fig. 5d–h).

To further evaluate possible connection between lipidome variation among regions and the human brain functionality, we assessed the relationship between the lipid abundance profiles and the functional architecture of information processing networks, specifically the signal processing hierarchy and resting-state synchronicity. Furthermore, lipids that are linked to these functional brain features display distinct

biochemical and cell-type-specific properties. For instance, lipids inversely correlated with the signal processing hierarchy tend to contain more DHA residues and are associated with inhibitory neurons. On the other hand, lipids that are positively correlated with the signal-processing hierarchy contain an excess of omega-3 polyunsaturated residues and are associated with astrocytes (Fig. 6f, g). These findings suggest the existence of gradients in neuronal and astrocyte density or subtype composition aligning with processing hierarchy. The existence of such anatomical patterns is supported by studies reporting anteroposterior neuronal density gradient in primates[51] and gradual changes in astrocytic representation between dorsoventral and rostrocaudal cortices in mice[52], both partially aligning with the signal processing architecture. Moreover, recent spatial transcriptomic analysis of the macaque brain has directly shown the relationship between variations in gene expression and the traditional anatomy and neurophysiology-based information flow hierarchy of the visual signal network[53]. It should be noted that the myelination signal does not determine the relationship between the composition of the lipidome and the signal processing hierarchy. This is evident from the fact that the *unexplained* lipid category, which is independent of myelin content, shows the strongest association with the brain's signal-processing architecture (Fig. 6c and Fig. 7).

In contrast to the myelin-independent nature of the lipid association with the signal processing hierarchy, our findings indicate that *myelin*+ lipids have the strongest association with the brain's resting-state network architecture. This suggests that there are differential properties of the lipidome related to axon myelination or differential representation of myelinated axon subtypes within the tracks

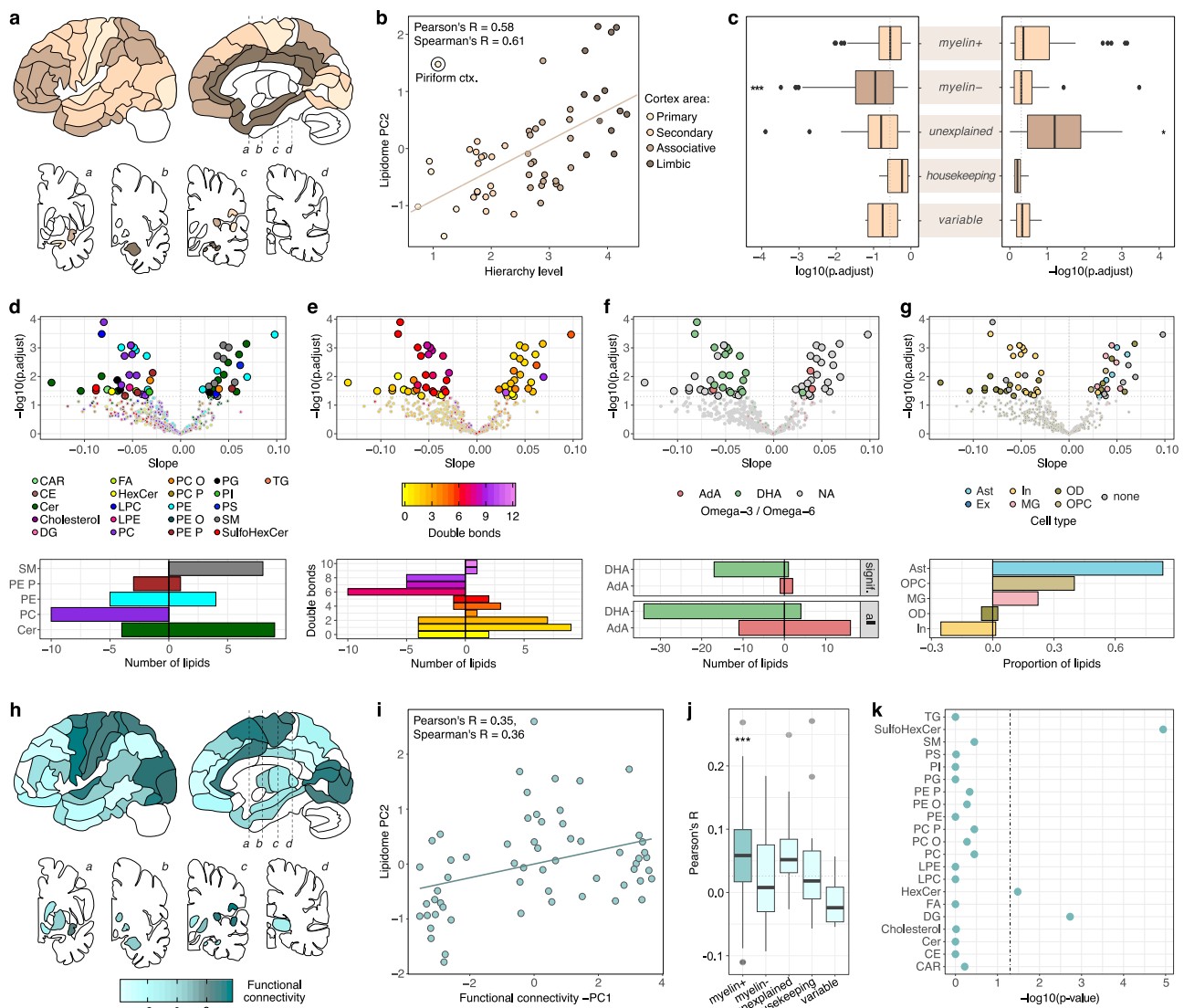

**Fig. 6 | Association between brain lipidome and the functional architecture.**
**a** Location of 56 brain regions assigned and colored according to their processing hierarchy (HR) within the brain. Brain images were adapted from ref. 1.
**b** Relationship between PC2 of the total variation of 419 HRMS lipids and HR in the human brain ($n = 4$ individuals here and in **c–k**). Circles represent brain regions. Circle marks the piriform cortex. **c** Significance of the relationship between lipid intensity and HR levels shown as correlation test $p$-value distribution based on all lipids within a category: negative correlations—left, positive—right. Asterisks indicate the significance of the difference among categories (one-sided Mann–Whitney $U$ test, $^{**}p < 0.01$; $^{*}p < 0.05$). **d–g** Volcano (top) and distribution (bottom) plots showing properties of lipids significantly correlated with HR: lipid class allocation (**d**), fatty acid residue unsaturation (**e**), omega-3/omega-6 prevalence based on

docosahexaenoic (DHA) or adrenic (AdA) acid occurrence (**f**), and cell type assignment (**g**). $P$-values were calculated via slope for a linear model, BH-corrected. **h** Location of 59 brain regions assigned and colored according to the inverted first principal component (−PC1) of their functional connectivity within the brain. Brain images were adapted from ref. 1. **i** Relationship between PC2 of the total variation of 419 HRMS lipids and −PC1 of FC. Circles represent brain regions. **j** Distributions of correlation coefficients between lipid intensity and FC in each lipid category. Asterisks indicate the significance of the difference among categories (one-sided Mann–Whitney $U$ test, $^{**}p < 0.01$; $^{*}p < 0.05$). **k** Significantly higher correlation with FC for individual lipid classes (one-sided Mann–Whitney $U$ test). Source data are provided as a Source Data file.

connecting regions within these networks. The relationship between the microstructure of cortical regions, as represented by intracortical myelin content, and functional network connectivity has indeed been demonstrated in humans using ultrahigh-resolution MRI[54], and aligns with earlier studies conducted in macaques and mice[55–57].

Our study has several limitations that should be considered. Firstly, our brain lipidome assessment was based on a small sample size of only four individuals. While this number aligns with previous transcriptome brain map studies[1–4], it may limit the generalizability of our findings. However, we were able to construct representative average lipid intensity estimates and the stability of these estimates was supported by assessing interindividual variation in humans and reproducibility in macaques, which served as a better-controlled sample group.

Additionally, our study design may have been underpowered to detect intensity alterations specific to a single brain region. This could potentially lead to the misassignment of certain lipids to either a housekeeping or variable category. Furthermore, our lipidome analysis did not include potentially informative biochemical lipid classes, such as cardiolipins – mitochondrial lipids known to have brain-specific variants[58], or gangliosides – glycosylated lipids abundant in the nervous system and implicated in neurodegenerative diseases[59–61]. In the association analysis of lipids and brain cell types, we were constrained by the indirect assignment of lipid profiles to cell types, which was based on the deconvolution of tissue sample composition using known cell type mRNA markers. Although this method is well established, it does have its limitations. In particular, while this

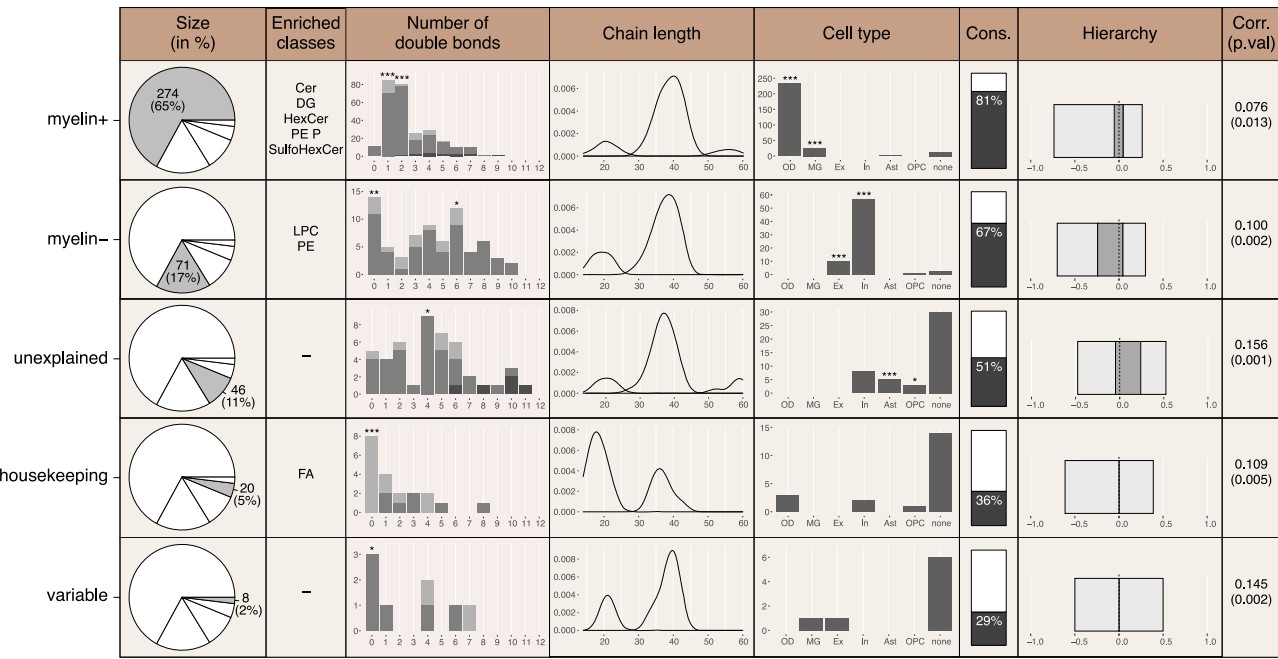

**Fig. 7 | Summary of lipid properties in the five profile-defined categories.** Columns, from left to right, show the following information: (1) The number and percentage of all detected lipids occupied by the category. (2) Significantly enriched lipid classes. (3) Bar plots depicting the distribution of the total number of double bonds within lipid fatty acid residues, color-coded based on their number (light gray−one, gray−two, deep gray−three). (4) Density plots showing the total lipid fatty acid chain length distribution, combining information for lipids with different numbers of fatty acid residues (left peak−one, middle peak−two, right peak−three). The counts for each group were normalized to maintain the relative heights of the peaks. (5) Numbers of lipids associated with the main brain cell types within each category. Lipid-cell type assignments were based on a Pearson correlation threshold of 0.5 between lipid intensity and mRNA expression profiles of cell-type marker genes across 35 brain regions. If correlations did not meet this threshold, the lipid was not assigned to a particular cell type (cell type = "none" including also ubiquitous lipids). (6) Conservation of lipid profiles between human ($n = 4$ individuals) and macaque ($n = 3$ animals) brains as the percentage of human lipids falling into the same category in macaques. (7) Correlation of lipid profiles with brain information processing hierarchy, with a darker shade of gray indicating lipids with correlation $p < 0.05$. (8) The strength and significance of the correlation between the lipid intensity matrices and the functional connectivity matrix. Asterisks, when present, indicate the enrichment of a specific lipid group within a given lipid category (hypergeometric test, BH-corrected, \*$p < 0.001$; \*\*$p < 0.01$; \*$p < 0.05$). Source data are provided as a Source Data file.

deconvolution procedure is effective at separating major cell types, it struggles with differentiation of cellular subtypes, which often display strongly correlated expression profiles. Moreover, our brain region selection was biased towards the neocortex, neglecting the detailed exploration of subcortical structures, connective axonal tracks, and infracortical white matter. This limits our understanding of the overall lipid composition of the brain. Additionally, our LC-MS analysis lacked information about the spatial distribution of lipids within the examined regions. To partially address this limitation, we utilized mass spectrometry imaging for selected regions.

Despite certain limitations, our study underscores the potential of lipid measurements to yield valuable insights into the structural and functional properties of the brain that may not be fully captured by other techniques. More broadly, we illustrate that the brain's lipidome content can serve as a bridge between the local molecular and cellular composition of individual regions and the large-scale structural and functional characteristics of the organ. This type of analysis could be particularly beneficial for studies examining molecular alterations in psychiatric and cognitive disorders that affect distributed brain structures. Several studies have identified lipids as promising molecular markers of brain disorders[62]. Consequently, our exploration of lipid variation across diverse regions in a healthy brain could provide a crucial baseline for future lipidome studies of brains affected by such disorders.

## Methods
### Brain tissue dissection
Post-mortem human and macaque brain samples were obtained from the National BioService, St. Petersburg, Russia. Informed consent for the use of human brain tissues for research was obtained from all donors or their next of kin by the tissue provider bank. The protocol was approved by the Skoltech Institutional Review Board. All human subjects were defined as healthy with respect to the sampled brain tissue by medical pathologists (Supplementary Table 1). Sex was determined based on mRNA expression of Y chromosome genes. All human subjects suffered sudden death with no prolonged agony state from causes not related to brain function. The macaques were not subjected to any pharmaceutical or immunologically related treatment anytime during their lifetime and were sacrificed for reasons others than participation in this study. They were part of a placebo group in a non-commercial study. The use and care of the non-human primates were carried out in accordance with Directive 2010/63/EU of the European Parliament and of the Council of 22 September 2010 on the protection of animals used for scientific purposes.

All post-mortem brain slices were stored at −80 °C. For sample dissection, The Atlas of the Human Brain[47] and The Rhesus Monkey Brain[63] were used to locate the areas of interest in the human and macaque brains, respectively. Frozen brain slices were atempered to −20 °C prior to dissection. Tissue pieces of approximately 10−15 mg in weight were cut out from selected brain areas using a metal scalpel. Dissected samples were then collected with tweezers, put into tubes, and immediately stored at −80 °C. All materials used during dissection (scalpels, tweezers, tubes) were sterile and chilled using dry ice before use. For each of the four human individuals, we dissected 75 samples representing anatomically defined brain structures (Supplementary Table 1). Similarly, for each of the three macaques, we dissected 38 samples from brain regions homologous to a subset of 75 human brain areas (Supplementary Table 1).

## Lipid extraction

Prior to extraction, 10–15 mg tissue pieces were dissected from the frozen tissue samples, weighed, and transferred to cooled 2 ml round bottom reinforced Precellys tubes (Bertin Technologies). The extraction buffer (methanol:methyl-tert-butyl-ether (1:3 (v/v)) was spiked with 0.5 μg/ml triacylglycerol (TAG 15:0/18:1-d7/15:0, Avanti Polar Lipids, 791648C), diacylglycerol (DAG 15:0/18:1-d7, Avanti Polar Lipids, 791647C), ceramide (Cer d18:1-d7/15:0, Avanti Polar Lipids, 860681), lysophosphatidylcholine (LPC 18:1-d7, Avanti Polar Lipids, 791643C), phosphatidylglycerol (PG 15:0/18:1-d7, Avanti Polar Lipids, 791640C), phosphatidylcholine (PC 15:0/18:1-d7, Avanti Polar Lipids, 791637C), and phosphatidylethanolamine (PE 15:0/18:1-d7, Avanti Lipids, 791638C) as internal standards. The buffer was prepared once for all samples and saturated with nitrogen prior to cooling to −20 °C. After stratified randomization, lipid extraction was performed in batches of 48 samples. After every 23rd sample, an extraction blank sample consisting of an empty tube without a tissue sample was inserted. Following the addition of 0.5 ml of −20 °C cold extraction buffer to each tube, including the extraction blanks, we performed homogenization of the tissue pieces using a Precellys Evolution Tissue Homogenizer (Bertin Technologies). After the addition of 0.5 ml extraction buffer, the homogenates were vortexed and then incubated for 30 min at 4 °C on an orbital shaker, followed by 10 min ultra-sonication in an ice-cooled sonication bath. The homogenate was transferred to 2 ml Eppendorf tubes followed by the addition of 700 μl of water:methanol mix (3:1 (v/v)). After brief vortexing, the homogenates were centrifuged for 5 min at 14.000 g to achieve separation of the hydrophobic and aqueous phases. Next, 540 μl of the upper phase containing hydrophobic compounds (predominantly lipids) was transferred to a 1.5 ml Eppendorf tube. The solvent was then evaporated using a Speed Vac concentrator at room temperature. For quality control (QC) samples, 5 μl of the upper phase containing lipid compounds of each analyzed sample was additionally collected and pooled. Lipid samples were then stored at −80 °C till the mass spectrometry analysis stage.

## MS sample preparation and measurements

Prior to mass spectrometry analysis, samples derived from one brain were grouped and randomized within the group for the processing order. All samples were run in one sequence without interruption. Batches of approximately 100 samples were prepared simultaneously. To resuspend the dried lipid pellets, 200 μl ice-cold acetonitrile:isopropanol (3:1 (v:v)) was added to each sample. After brief rigorous vortexing, the samples were shaken for 10 min, sonicated in an ice-cooled sonication bath for 10 min, and centrifuged for 5 min at 14.000g. The supernatant was transferred to a fresh 1.7 ml Eppendorf tube. For mass spectrometry analysis, 25 μl of lipid sample was transferred to a 2 ml autosampler vial and diluted 1:15 with 350 μl acetonitrile:isopropanol (3:1 (v:v)). Then 3 μl per diluted sample were separated on a Reversed Phase Bridged Ethyl Hybrid (BEH) C8 column reverse (100 mm × 2.1 mm, containing 1.7 μm diameter particles, Waters) coupled to a Vanguard precolumn with the same dimensions, using a Waters Acquity UPLC system (Waters, Manchester, UK). The mobile phases used for the chromatographic separation included: water containing 10 mM ammonium acetate, 0.1% formic acid (Buffer A), and acetonitrile:isopropanol (7:3 (v:v)) containing 10 mM ammonium acetate, 0.1% formic acid (Buffer B). The gradient separation protocol included the following steps: 1 min 55% Buffer B, 3 min linear gradient from 55% to 80% Buffer B, 8 min linear gradient from 80% Buffer B to 85% Buffer B, and 3 min linear gradient from 85% Buffer B to 100% Buffer B. After 4.5 min washing with 100% Buffer B the column was re-equilibrated with 55 % Buffer B for 4.5 min. The flow rate was set to 400 μl/min.

The mass spectra were acquired in positive and negative modes using a heated electrospray ionization source in combination with a Bruker Impact II QTOF (quadrupole-Time-of-Flight) mass spectrometer (Bruker Daltonics, Bremen, Germany). Negative ion mode samples were run after completion of the positive ion mode run using a 2 μl injection of non-diluted lipid samples. The spectra were recorded using full scan mode, covering a mass range from 50–1200 m/z. MS settings in positive polarity were as follows: capillary voltage 4000 V, nebulizer 2 Bar, dry gas 6.0 l/min, dry temperature 180 °C. MS settings for negative polarity were as follows: capillary voltage 4000 V, nebulizer 2 Bar, dry gas 6.0 l/min, dry temperature 200 °C. For quality control (QC), a pooled sample of all the lipid extract was prepared in and injected six times before initiating the runs to condition the column, at least five times after each sub-cohort, and after the completion of the runs. In addition, the QC sample was injected after every 10 regular brain lipid samples to assess instrument stability and analyte reproducibility. Four blank samples consisting of acetonitrile:isopropanol (3:1 (v:v)) were measured at the start of the run. The extraction blank samples were queued at the end of the run preceded by eight injections of pure acetonitrile.

## Data extraction and preprocessing

After the acquisition, Bruker raw data output (.d files) were automatically calibrated by internal calibration followed by lock mass calibration and converted into mzXML format using a custom DataAnalysis script (Bruker, Version 4.3). Obtained mzXML files were then transferred to XCMS software using the *xcms* package for R version 3.8.2[64].

Mass spectrometric peaks falsely duplicated during the XCMS peak merging procedure were identified using a 10 ppm mass threshold within 1 s retention time difference. Peaks detected during the first minute of the run (retention time <1 min) and after retention time = 18.3 min were excluded from further analysis. The 'fillpeaks' procedure implemented in the *xcms* package was used for missing value imputation. The lipid intensity values missing after the 'fillpeaks' procedure were filled by random sampling from a normal distribution with the mean equaling the median of minimal intensity values of detected lipid peaks and the standard deviation equaling the 16th percentile of this distribution.

## Lipid Identification

To annotate mass spectrometry peaks, we considered 23 lipid classes referenced by the literature with respect to the primate brains (PE, PC, PE O, PC O, PG, PS, PA, PI, LPC, LPE, LPC O, LPE O, LPS, LPA, Cer, SM, HexCer, GM3, CE, SulfoHexCer, TAG, DAG, MAG and cholesterol) and searched for the corresponding lipid species in our data. Lipid classes were defined according to corresponding LIPID MAPS[38] subclasses, except for several special cases: 1) all free fatty acids were merged in one class; 2) Cer class also included dihydroceramides and phytoceramides subclasses from LIPID MAPS; 3) HexCer class contained both galactosylceramides and glucosylceramides. Mass spectrometry peak annotation included the following steps. First, we constructed a custom database of mass-to-charge values, varying the chain lengths and the number of double bonds, based on selected 23 lipid classes, to match with the mass spectrometry peaks of our dataset. Next, we matched the mass-to-charge values of all the lipid species of each lipid class with a fixed adduct using a 10 ppm threshold. We kept only those lipid classes for which a visually distinguishable 'grid' on the mass-to-charge versus retention time scatterplot could be found, filtering out the mass spectrometry peaks with retention times that did not match the grid-like pattern, additionally using internal standard retention time as an anchor point, when available. This procedure was first performed in each ionization mode with a class-specific adduct of choice. Putative annotations were then verified by cross-validation between two ionization modes (when available for the lipid class) by matching the retention times of the peaks between positive and negative ionization modes. We further validated putative annotations

based on molecular ion masses and retention times using fragment ions information obtained in a tandem mass-spectrometry experiment. The following statistical analysis included lipids with confirmed lipid class annotation. Fragment ions data also yielded more detailed information about the fatty acid composition of the lipids. Fragmentation was performed on pooled samples, representing mixed aliquots of brain regions including both white and gray matter components. The brain regions with the highest average intensity of a given target lipid class were chosen for this analysis.

For the fragment ion analysis, conditions of chromatographic separation were maintained the same as described in the section "MS sample preparation and measurements" and mass spectra acquisition was performed on a hybrid QExactive instrument in data-dependent (DDA) mode. Spectra were recorded separately in positive and negative ionization modes. In each mode, resulting spectra were obtained by averaging three fragmentation spectra at different collision energies. Mass spectrometer was equipped with heated electrospray ionization (H-ESI) and tune parameters were set as follows: capillary temperature: 320 °C; aux gas heater temperature: 350 °C; capillary voltage: 4.5 kV (−3.5 kV); S-lens RF level: 60; sheath gas flow rate ($N_2$): 45 arbitrary units (a.u.); auxiliary gas flow rate ($N_2$): 20 a.u., sweep gas flow rate ($N_2$): 4 a.u. Probe parameters were identical for both polarities. The operating parameters of the mass spectrometer for full scan mode were set as listed: resolution: 70000 at $m/z$ 200, automatic gain control (AGC target): 5e5, maximum injection time (IT): 50 ms, scan range: 200 to 2000 Da. For DDA mode: resolution 17500 at $m/z$ 200, AGC: 2e4, IT: 100 ms, mass isolation window: 1.2 Da, retention time window width: expected time ± 1 minute, stepped normalized collision energy: 15, 25, 30%, dynamic exclusion 12 s, inclusion: on, customize tolerances: 10 ppm. The spectra were recorded in the profile mode.

The lipid formulas proposed in this work are based on the match of the precursor and fragment ion masses with the theoretical ones with an accuracy of less than 10 and 15 ppm, respectively, and expected chromatographic behavior (increasing retention time for series of homologs and decreasing with the addition of double bonds). Acquired MS2 spectra were manually curated, and lipid class assignments were performed based on their distinctive fragmentation behavior. Lipids were attributed to phospholipids by the presence of fragment (or neutral loss) signals related to the polar head group, to glycerolipids – by neutral loss of ammonia for the precursor ion, and to sphingolipids – by characteristic ions of the sphingoid base. The fatty acid composition for phospholipids was interpreted by the presence of deprotonated fatty acid ions in MS2 spectra, for glycerolipids by the neutral loss of fatty acids as ammonium adducts, and for sphingolipids by the difference in the masses of precursor ions and ions of sphingoid bases.

Our lipid structure elucidation refers to "FA acyl/alkyl level" according to the proposed lipid nomenclature[65] or to "putatively characterized compound classes (level 2)" according to the metabolomics standard initiative[66]. It implies that sn-attachments of FA, positions of double bounds, and stereochemistry are not declared. The retention behavior of seven lipid classes was confirmed by class-specific isotopically labeled standard (see section "Lipid extraction"). Since ether-linked lipids (plasmanyl- and plasmenyl-phospholipids) cannot be distinguished by class-specific fragments, we utilized their chromatographic behavior for annotation. As we observed sufficient separation between closely eluted isomeric species differing in double bond position (e.g., PC-P 36:2 and PC-O 36:3), both species were reported. The elution order was validated using the authentic standard of PC-P 18:0/18:1(9Z) (Avanti Polar Lipids, 852467 C).

Lipids were attributed to phospholipids by the presence of fragment or neutral loss (NL) signals related to the polar head group*, to glycerolipids by neutral loss of ammonia for the precursor ion**, and to sphingolipids by characteristic ions of the sphingoid base***. The fatty acid composition for phospholipids was interpreted by the presence of

deprotonated fatty acid ions in MS2 spectra, for glycerolipids by the neutral loss of fatty acids as ammonium adducts, and for sphingolipids by the difference in the masses of precursor ions and ions of sphingoid bases. Free fatty acids were annotated based on the exact mass of deprotonated molecules in MS1 scanning mode.

\*
positive polarity: 184.0739 (C5H15NO4P + ) – LPC, PC, PC-O, PC-P; 18.0106 (NL H2O) – LPC; 141.0191 (NL C2H8NO4P) – LPE, PE, PE-P, PE-O; loss of the glycerol moiety as a neutral olefin (unique for each plasmalogen sn-1 group) - PE-P; 185.0089 (NL C3H8NO6P) – PS; 172.0137 (NL C3H9O6P) , 190.0475 (C3H13NO6P + ) – PG;

negative polarity: 168.0458 (C4H11O4NP-), 224.0688 (C7H15O5NP-), 60.0211(NL C2H4O2), 74.03678 (NL C3H6O2) - LPC, PC, PC-O, PC-P; 140.0113 (C2H7O4NP-), 196.0375 (C5H11O5NP-) - LPE, PE, PE-P, PE-O; 241.0113 (C6H10O9P-), 259.0226 (C6H12O10P-), 223.0008 (C6H8O8P-)– PI; 87.0320 (NL C3H5NO2) – PS;

\*\*
positive polarity: 17.026549 (NL NH3) - TAG, 35.0371 (NL NH3 + H2O), 18.0106 (NL H2O) – DAG. Fatty acid composition was established by NL of a neutral carboxylic acid and neutral ammonia together.

\*\*\*
positive polarity: 184.0739 (C5H15NO4P+) – SM; 180.0634 (NL C6H12O6) – HexCer; the characteristic product ions of the sphingoid bases for Cer, HexCer and sulfotides: 250.2534 (C17H32N+- d17:1), 264.2691 (C18H34N+- d18:1), 262.2534 (C18H32N+- d18:2), 292.3004 (C20H38N+- d20:1), 268.3004 (C18H36N+- m18:0), 266.2847 (C18H34N+- m18:1), 296.3317 (C20H38N+- m20:0), 294.3160 (C20H36N+- m20:1), 268.2640 (C17H34NO+), 250.2534 (C17H32N+- t17:0), 282.2796 (C18H36NO+), 264.2691 (C18H34N+- t18:0). Due to in-source water loss from protonated Cer and HexCer molecules, monitoring of [M+nxH2O]+ ions and [M-nxH2O]+ ms1-ions along with [M + H]+ ions was performed to distinguish between sphingolipid bases with different number for hydroxyl groups. The following fragment ions were used to establish sphingoid base composition of Cer, HexCer, sulfotides: [base+H-H2O]+ for mCer, [base+H-2xH2O]+ for dCer, [base+H-2xH2O]+ and [base+H-3xH2O]+ for tCer.

negative polarity: 96.960 (HSO4-) – sulfatides.
Other lipids:
positive polarity: 85.028 (C4H5O2+) - CAR, 369.352 (C27H45+) – Cholesterol, CE.

## Targeted MS measurements using LC-MS/MS (MRM)

For the targeted analysis, we selected 341 transitions from the total list of signals obtained in an untargeted experiment. Depending on the lipid class identity, we prepared two sample dilutions. For the measurement of SM, PE, PE-O, PE-P, PC, PC-O, PC-P, LPC, and LPC transitions, samples were diluted 1:100 with acetonitrile:isopropanol (3:1 (v:v)). For the measurement of PG, PI, PS, DAG, Cer, HexCer, SHexCer, and CE transitions, samples were diluted 1:10 with acetonitrile:isopropanol (3:1 (v:v)). 2 μl per diluted sample were separated on an EclipsePlus C18 RRHD column (2.1 × 50 mm, 1.8μm, Agilent Technologies), using an Agilent 1290 Infinity Binary Pump. The mobile phases used for the chromatographic separation were acetonitrile:water (4:6 (v:v)) containing 10 mM ammonium formate (Buffer A) and acetonitrile:isopropanol (1:9 (v:v)) containing 10 mM ammonium formate (Buffer B). The gradient separation was: 2 min 60 % B, 7 min 100 % B, 9 min 100 % B, 9.01 min 20 % B, 10.80 min 20 % B. The flow rate was set to 400 μl/min. MRM were acquired in positive and negative modes using a heated electrospray ionization source with an Agilent jet stream in combination with an Agilent 6495 Triple Quadrupole mass spectrometer (Agilent Technologies, USA). All samples were measured in one uninterrupted run including positive ion mode for all transitions and negative ion mode for PI transitions. MS settings: gas temperature 120 °C, gas flow 11 l/min, nebulizer 40 psi, SheathGasHeater 300, SheathGasFlow 10, capillary voltage 3500 V for positive and 3000 V for negative polarity, VCharging

500 for positive and 1500 for negative polarity. QC dilution series (DiQC) were run at the beginning (0.0625, 0.125, 0.25, 0.5, 1) and at the end of the queue (1, 0.5, 0.25, 0.125, 0.0625). Pooled QC sample was run after every 10th brain sample, followed by a blank sample containing only solvent; every second blank was followed by an extraction blank; every third QC was preceded by the long-term reference sample (LTR, extract of the plasma lipids).

## MRM data processing

All transitions were integrated with MassHunter Quantitative Analysis software version B.08. Peaks with intensity levels in blanks exceeding 0.1 of the intensity levels in real samples were excluded; peaks with the coefficient of variation in QC samples above 25% were excluded; peaks with the correlation coefficient between the DiQC series below 0.9 were excluded. Out of 341 transitions, 169 passed all filters and were considered for subsequent analysis.

## Peaks filtration and preprocessing

To account for extraction and technical noise, we applied filtering procedures to the resulting lipid compound target list. First, a blank samples filter was applied: only the features with mean intensity in samples being at least two times higher than in blanks were selected for further analysis. Second, a variance filter was applied: for each peak coefficient of variation was calculated across QC samples in each batch, and only peaks with median CoV lower than 0.25 were selected.

Lipid intensities, defined as areas under mass spectrometric peaks, were $\log_{10}$-transformed, and all resulting lipid intensities were normalized on the median value of standards in a sample and the wet weight of the sample. For normalization wet weights and standard intensities were $\log_{10}$-transformed, then the difference of sample values from the mean of a parameter was subtracted. Resulting lipid intensity values were further normalized using the mean intensity of this lipid calculated within each individual's brain to adjust for inter-individual variability.

## RNA expression data analysis

RNA sequencing was performed for 35 human brain regions representing a subset of 75 brain regions used in lipid analysis in the same four human individuals. Data from 33 brain regions were previously published[1], while data for an additional two brain regions generated in the same experiment was not previously released. The experiment was performed as described in ref. 1. Briefly, total RNA was isolated using Direct-zol-96 RNA (Zymo Research) from 10 mg of the frozen tissue per sample. Sequencing libraries were prepared with NEBNext Ultra II RNA Library Prep Kit (New England Biolabs) according to the manufacturer's instructions. Libraries were then sequenced on the Illumina HiSeq 4000 system using the 150-bp paired-end sequencing protocol.

Quality assessment of raw data with the fastQC[67] revealed that many reads have Illumina universal adapters at the end. To trim low-quality bases and remove adapter sequences we used the trimmomatic tool[68] with the following parameters: "PE -phred33 ILLUMINACLIP:all.fa:2:30:10:2:true SLIDINGWINDOW:4:15 LEADING:3 TRAILING:3 MINLEN:20", we used the union of all adapter sequences provided with trimmomatic and, additionally, we included the sequence of Illumina universal adapter found by the fastQC (AGATCGGAAGAG) for palindrome clipping (see trimmomatic manual). Then reads were mapped to corresponding genomes using hisat2[69] with the following parameters: "--no-softclip --max-intronlen 1000000 -k 20". Gene expression values (Transcripts Per Kilobase Million, TPM) were estimated using the stringtie[70] with the following parameters: "-e -G reference.gtf -B out -A out.tab". Genome sequences (GRCh38, Mmul_8.0.1, panpan1.1 and Pan_tro_3.0), gene annotations used in the analysis, and a table of orthologous genes for all four species were obtained from Ensembl v91[71].

Acquired sequence read counts were next transformed to TPMs, and only genes with TPM > 1 were selected for further analysis. All TPM values were $\log_{10}$-transformed and normalized using the median value of each sample. Only protein-coding genes were used in the analysis. As in the case of lipids, the expression values of each gene were normalized by the mean expression level of this gene within each brain.

## Region clustering

Prior to hierarchical clustering, we calculated the mean lipid intensity profiles across all individuals spanning all analyzed regions. We used Euclidean distance as a dissimilarity measure and a "complete linkage" method for hierarchical clustering implemented in R *hclust()* function[72]. We used the tSNE method for lipidome-based sample similarity visualization. We used ANOVA method to select lipids with significantly different intensity levels in different regions with the model: *lipid ~ region*. To compare spatial distances between regions to lipid intensity and gene expression distances, Euclidean distance matrices were calculated for each dataset. Next, Pearson's R was calculated for each region's vector of distances and the corresponding vectors of distances in two other datasets: lipidome and transcriptome. For this analysis, we focused on 19 neocortical regions contained in both lipidome and transcriptome datasets.

## sMRI and rs-fMRI data

The anatomical positions of dissected samples were aligned to the standardized MNI coordinate space of human and macaque brains based on sample preparation notes and images. We used preprocessed sMRI data recorded from alive individuals from Human Connectome Project 1200 release from[7]. The myelin content of each region was evaluated as the mean intensity of T1w/T2w signal in a cube $9 \times 9$ mm centered around the coordinate of the center for each sample. Similar data from macaque was taken from[73] and preprocessed in the same way. For the functional connectivity analysis, data extraction regions of interest were matched to regions from ref. 48, covering 59 of the 75 analyzed regions. In the case of multiple matching between the regions, the mean value of multiple matched regions was taken.

## Lipid clustering

The categorical classification of lipids based on their profiles across 75 regions was performed stepwise. First, the ANOVA method was applied to select lipids with significantly different intensity levels among brain regions (BH-corrected *p*-value < 0.01). Second, a linear model was built for each ANOVA-selected lipid comparing their intensity with the calculated myelin content of the regions: lipids with significant positive correlation (BH-corrected p-value for linear model <0.05) were classified as *myelin+* lipids; lipids with significant negative correlation (BH-corrected p-value for linear model <0.05) were classified as *myelin-* lipids; the remaining lipids were marked as *unexplained*. Finally, lipids showing no significant intensity differences among brain regions were divided into two groups based on their within-region variability. For this purpose, the Gaussian component of the within-regions variability was extracted, and the threshold was set at $\mu + \sigma$. Lipids with within-region intensity variability falling below the threshold (inside the Gaussian bell curve) were classified as *housekeeping*, and the remaining lipids were classified as *variable*. Lipid class, number of double bonds, and chain length enrichment of lipid categories were evaluated using a hypergeometric test, implemented in R as *phyper()* function. The hypergeometric test is employed to assess the statistical significance of drawing a specific number of successes from a population with a known quantity of successes. For instance, in the enrichment analysis of lipid classes, all measured lipids serve as the population. This population contains a specific number of lipids in a particular class, representing the possible number of successes. The lipids within a specific profile category represent the lipids we draw.

## Gene ontology (GO) enrichment analysis

Gene Ontology (GO) enrichment analysis was conducted for gene sets correlated with the mean lipid cluster profile drawn across 69 regions (based on Allen Brain data) with a coefficient greater than 0.5. For each category of GO terms (cellular compartment, biological processes, and molecular function) the enrichment analysis was conducted using the function *enrichGO()* from *ClusterProfiler* package[74] for BF (biological process), CC (cellular compartment), and MF (molecular function) subontologies. All genes detected in the experiment were used as background. *Simplify* method was applied to selected categories to remove highly intersecting terms (threshold = 0.5), and a significance threshold for BH-adjusted enrichment p-values was set to 0.001.

## Comparison of human data to macaque data

To analyze the macaque brain lipidome dataset, we applied the lipid annotation procedure used for the human samples, including LC-MS/MS validation, without modifications. All lipids annotated in macaque data and matching the list of annotated human lipid compounds were selected for further analysis. Lipid clustering and classification procedures were identical to the ones used for the human data but relied on macaque sMRI data calculated for the 38 analyzed regions. Differential stability scores were calculated, as described in ref. 4 and cross-correlation coefficients between individuals were averaged for each lipid. To evaluate the significance of the correlation between corresponding lipids in the human and macaque datasets, for each lipid we permuted sample labels and compared Pearson's R between two datasets for true and permuted labels by paired Wilcoxon test. To test the stability of the lipid classification procedure, all cluster assignments were classified either as "good" (the same cluster assignment in human and macaque datasets), "neutral" (assignment to a non-matching cluster with no significant difference between regions compared to the matching one in at least one dataset) and "bad" (assignment in different clusters with significant difference between regions in human and macaque datasets).

## Cell type analysis

To find an association between lipid intensities distribution across regions and the cellular composition of the brain we used mRNA data derived from 35 regions, representing a subset of 75 regions analyzed for their lipid content. The same tissue samples were used both for mRNA and lipid measurements. We used a manually selected from the literature well-characterized set of cell type markers corresponding to the six main brain cell types[1]: oligodendrocytes (OD), microglia (MG), astrocytes (Ast), oligodendrocyte progenitor cells (OPCs), inhibitory (In) and excitatory (Ex) neurons (Fig. 3f). Lipid assignment to cell types was based on the correlation between lipid intensity profiles constructed across 35 brain regions and mRNA expression profiles of cell type marker genes constructed across the same 35 regions. A lipid was assigned to a cell type based on the highest Pearson correlation with expression marker profiles and exceeding the correlation coefficient threshold = 0.5. If no correlations to the expression profiles of any cell type markers passed the threshold, the corresponding lipid was marked as non-assigned to a cell type (cell type = "none"). Cell type markers enrichment analysis of lipid categories was carried out using the hypergeometric test. The test was conducted for each cell type and lipid category using all lipids as a background comparison population. Comparison of lipid intensities before and after the cell sorting experiment was carried out using a list of top-20 lipid cell type markers showing the highest correlation with expression markers of a given cell type (or less, if a total number of lipid markers was included less than 20 lipids for a given cell type). In (inhibitory neuron) and Ex (excitatory neuron) markers were merged together in a group of "neuronal" markers, while OD, OPC, Ast, and MG lipid markers were merged as a "non-neuronal" group.

## Fluidity estimation

Two measures of a membrane fluidity computation were used for each lipid: mean number of double bonds per carbon chain and mean chain length. For this purpose, the total number of double bonds and the total number of carbon atoms were divided by the number of carbon chain residues typical for the class. For this type of analysis, cholesterol was not considered. Each value was normalized to range between 0 and 1 by subtracting the minimal value (among all lipids) and dividing by the difference between the maximal and minimal value (among all lipids). Since the membrane fluidity gets higher with an increasing number of double bonds and gets lower with increasing carbon chain length, the normalized chain length was subtracted from 1 and averaged with the normalized number of double bonds. The resulting value reflected the relative normalized influence on membrane fluidity provided by the given lipid.

## FACS experiment for rodent brain

For the lipidome analysis of sorted cell populations, we used Tg(Thy1-COP4/EYFP)9Gfng (Thy1-ChR2-YFP) transgenic mice (*n* = 2 animals). Expression of YFP in this line was reported in layer 5 cortical neurons, CA1 and CA3 pyramidal neurons of the hippocampus, cerebellar mossy fibers, neurons in the thalamus, midbrain, and brainstem, and the olfactory bulb mitral cells[75]. Adult female mice were anesthetized and decapitated following the standard, ethically acceptable procedure. The protocol was approved by the Skoltech Institutional Review Board in accordance with the guidelines on the ethical use of animals. All possible efforts were made to minimize animal suffering, and to reduce the number of animals used per condition by calculating the necessary sample size before performing the experiments. Mice were housed in standard breeding cages at constant temperature (22 ± 1 °C) and relative humidity (50%), with a 12:12 h light:dark cycle. Intact brains were isolated from the sacrificed animals within 3 min postmortem, minced by two cold razor blades on ice cold glass plate and placed in an ice-cold solution of zinc fixative (0.1 M Tris-HCl, pH = 6.5, 0.5% ZnCl2, 0.5% zinc acetate, 0.05% CaCl2, final pH = 6.3[76]) in at least 10 × volume at 4 °C for 2 h. Tissues were washed 3 times in PBS (20 min/wash). Fixed and washed tissue was dissociated by Medimax machine with Medicons-P disposable disaggregator with about 50-100 μm separator screen for 10 s in 1 ml of PBS. The suspension was filtered through SmartStrainers 70 μm filters. Filtered cells were spun at 250 g for 3 min in a swinging bucket centrifuge at 4 °C. Supernatant was carefully removed, the cells were resuspended in 200 μL of PBS, and stored at 4 °C until sorting.

FACS experiments were performed using a FACSAria SORP instrument (BD Biosciences). The samples were additionally stained with Hoechst33342 to distinguish the nuclei-containing cell bodies. The excitation and emission wavelengths for YFP and Hoechst33342 detection were as follows: Ex. 488 nm, Em. 505LP + 525/20 nm BP and Ex 407 nm, Em. 450/50 nm BP. Gates were set on two populations of interest: non-neuronal nuclei-containing parts (Hoechst33342-positive, YFP-negative) and neuronal YFP-containing parts without nuclei (Hoechst33342-negative, YFP-positive). Sorting was conducted in Purity mode, using an 85 μm nozzle with the corresponding pressure settings. Sorted specimens were frozen at dry ice immediately after the experiment and kept at −80 °C for further lipid analysis.

## Comparison to published data

To compare our human HRMS lipid intensity dataset to published mouse brain lipidome atlas data[21] and murine cell type-specific primary culture lipid data[20], we downloaded the original published raw intensities and preprocessed them using our human HRMS data processing pipeline with no modifications. We then calculated Pearson's correlation coefficients based on the intensity measurement vectors of lipid compounds matched by annotation between the datasets in the eight regions present both in human and mouse data (both murine datasets

had eight regions in common with our human data). Median diagonal and off-diagonal values in matrices of Pearson's R between two datasets were calculated in 100 region label permutations to test the non-random level of correlation between the two datasets. The resulting distributions were compared using paired Wilcoxon test. For each cell type, raw intensities equal to zero were replaced by random noise values, as described in the section *Data extraction and preprocessing* of the Methods, and all values were normalized using total signal intensity within each sample. For the cell type comparison between human and mouse datasets, the mean intensity for each cell type was calculated for each lipid class. The standard deviation was calculated across all mouse individuals. For the human dataset, the proportion of lipids from a given lipid class serving as cell-type markers was taken for comparison.

To compare human white matter lipidome data to HRMS measurements, raw data from ref. 18 was upper-quartile normalized for each sample and $\log_{10}$-transformed. The difference in the median intensity of white matter and gray matter samples was calculated for each peak. The statistical significance of this difference was assessed using *t* test and *p*-values were BH-corrected for multiple testing. Lipid peaks were matched between the two datasets based on their retention time value and mass-to-charge ratio dataset within the ±30 seconds and ±10 ppm windows. Prior to this procedure retention times were aligned between the two datasets based on a subset of lipid features matching within a more relaxed time window of ±120 seconds. Retention time correction was performed using the svm() function from the e1071 R package. For lipids passing the adjusted p-value threshold of 0.05 and fold-change threshold of two, an association between the sign of the difference and the myelin+/myelin- category was tested using Fisher exact test.

## MALDI imaging

MALDI imaging was conducted on the fresh frozen sections from the posterior part of superior temporal gyrus dissected from postmortem brain samples of 3 healthy donors. Fresh frozen brain tissue was sectioned into 10 μm slices using a cryostat Leica CM1950 at −18 °C. Slices of tissue sections were thaw-mounted on ITO slides (15Ω, HST, US). Sections were dried in the vacuum chamber for 2–3 h, and matrix was applied by spraying. α-Cyano-4-hydroxycinnamic acid (CHCA) was used as the MALDI matrix. A saturated solution of CHCA at 5 mg/mL−1 in 50/50 water/acetonitrile with 0.1% TFA was diluted two times. This solution was applied using an airbrush (Iwata Micron CM-B2) for 2 s and left to dry for 2.5 min, the cycle was performed 20 times. Neighboring sections were mounted on SuperFrost slides (Thermo Scientific, US) for subsequent histological staining, and stored at −80°C until processing. Histology of white and gray matter on the brain sections was revealed by luxol fast blue staining (blue color for lipid-rich compartments) and by eosin (pink color for protein-rich cytoplasm). Brain sections were briefly de-fated to increase dye penetration: placed gradually in ethyl alcohol (50%, 75%, 95% and 100%) and back to 95% for 1 min in each solution. Then sections were left in 0,1% luxol fast blue solution in ethyl alcohol with 0,5% glacial acetic acid in 56 C oven for 12–14 h. Excess stain was rinsed off with 95% ethyl alcohol and then in distilled water. Staining was differentiated in the 0,05% lithium carbonate solution in water for 30 seconds and rinsed in distilled water several times with constant control by microscopic examination if gray matter is clear and white matter sharply defined. Then sections were counterstained with 1% eosin solution in water for 30-40 seconds, rinsed in distilled water, dehydrated briefly in IsoPrep (BioVitrum), cleared in Bio Clear (Bio-Optica) and coversliped with Bio Mount HM (Bio-Optica). Histology images were acquired with Zeiss Axio.Observer.Z1 transmitted light microscope system.

MALDI images were obtained using a modified MALDI-Orbitrap mass spectrometer[77] (Thermo Scientific Q-Exactive orbitrap with MALDI/ESI Injector from Spectroglyph, LLC) equipped with a 355 nm

Nd:YAG Laser Garnet (Laser-export. Co. Ltd, Moscow, Russia). For positive ion induction, the laser power was set to a 20 J repetition rate of 1.7 kHz. The distance between the sample on a coordinate table and ion funnel was 5 cm. Produced ions were captured by the ion funnel and transferred to QExactive Orbitrap mass spectrometer (Thermo). Mass spectra were obtained in the mass range of *m/z* 500–1000 and a mass resolution was 140,000. The tissue region to be imaged and the raster step size were controlled using the Spectroglyph MALDI Injector Software. To generate images, the spectra were collected at 40-μm intervals in both the x and y dimensions across the surface of sample[77]. Ion images were generated from raw files (obtained from Orbitrap tune software) and coordinate files (obtained from MALDI Injector Software) by Image Insight software from Spectroglyph LLC (www.spectroglyph.com). MALDI raw mass spectra were converted to *.ibd and *.imzML formats using Spectroglyph software (https://spectroglyph.com/), with the background noise threshold set to zero. All further processing was done using Cardinal 2.8.0, an R package designed for mass spectrometry imaging data analysis[78].

For image analysis, duplicated coordinates were removed from converted files, then spectra were normalized by the total ion current. We performed peak picking based on a signal-to-noise ratio threshold equal to three to select peak centers for the downstream analysis. Signal to noise ratio was calculated based on the difference between the mean peak height in a window of a predefined size and the mean height in the window of the manually selected flat part of the spectrum[78]. Peaks present in less than 7% of the sample spectra were removed from further analysis.

Peaks originating from the glass slide surface not covered by tissue and uninformative parts of the spectra containing no biologically relevant peaks were removed from each sample spectral data. To do so, the sample image area was divided into two parts using the spatial k-means algorithm; one of the clusters corresponding to the tissue sample and the other – to the sample-free matrix-covered surface of the glass slide. The mapping of the clusters to the sample and the surrounding area was manually curated by visual inspection of the slides. Mean feature intensities were calculated for the two clusters and only peaks with 1.5 times greater mean intensities within the sample cluster compared to the surrounding sample-free area were kept. After this filtration step, all sample spectra were aligned to a spectrum with the largest number of detected peaks.

For the clustering of pixels within the sample area, the first principal component of the ion intensity matrix was used. The resulting clusters represented white and gray matter and were annotated based on histological staining of the adjacent sample sections. $\log_{10}$-transformed difference between median intensities of non-zero values in the white matter and gray matter clusters was calculated for each peak in each sample. To match HRMS annotated peaks to MALDI data, only peaks with 99% quantile intensities greater than 1000 were selected. Peaks were considered matched between the datasets if the mass difference between HRMS and MALDI was below 10 ppm. Non-uniquely matched peaks were removed from further analysis.

## ToF-SIMS experiment

Brain samples were sectioned and processed as described above in the *MALDI imaging* section. ToF-SIMS measurements were performed using a ToF-SIMS 5 instrument (ION-TOF Gmbh, Germany) equipped with 30 keV Bi3+ primary ions. Mass spectra were recorded in spectroscopy mode with pixel size ~ 5 μm. First mass spectra and chemical maps were acquired for positive secondary ions and then for negative. The region of interest and analysis area was 1.8 × 1.8 mm (384 × 384 pixels) while primary ion dose density (PIDD) did not exceed $3.2 \times 10^{10}$ ions/cm2. Stage scan was utilized for all experiments. A low-energy electron flood gun was activated to avoid charging effects. Ion yields were calculated as the intensity of the corresponding peak of interest normalized to the total ion count amount.

In the standards validation experiment, droplets of lipids standard were applied and dried on ITO coated slide type II 1.1 mm thick (P/N PL-IC-000010-P100, Hudson Surface Technology inc.). Eleven lipid standards manufactured by Avanti Polar lipids used in the experiment included: PE(17:0/17:0) (P/N 830756X), PG(17:0/17:0) (P/N 830456X), PI(8:0/8:0) (P/N 850181P), 17:0 SM (P/N 860585P), Lyso PC (17:0/0:0) (P/N 855676C), Lyso PE(14:0/0:0) (P/N 856735X), C17 Glucosyl(ß) Ceramide (d18:1/17:0) (P/N 860569P), Cer(d18:1-d7/15:0) (P/N 860681), d7-cholesterol (P/N 70004), C17 Ceramide (d18:1/17:0) (P/N 860517P) and mix of Brain Sulfatides (P/N 131305P-5mg). The concentration of all standards was 1 mole/ml, and droplets of 1–2 µl were dried at room temperature. ToF-SIMS measurement was performed as described above.

Five ions, validated by standards and previously reported in literature[79–81] were selected for comparison to HRMS data. Ion images of four samples were divided into white matter and gray matter spatial clusters based on histological staining of adjacent sections. For each sample mean intensity through all pixels was calculated for each cluster and the $\log_{10}$-transformed difference between white matter and gray matter clusters was calculated for each ion. For groups of lipid classes, corresponding to selected characteristic ions, the difference between proportions of lipids from a tested group contained within *myelin+* and *myelin-* clusters based on HRMS data were normalized by the mean value of two fractions.

### Functional brain activity analysis

For comparison of lipid intensity profiles to the hierarchical organization of the human brain, we build a linear model based on a subset of neocortical regions corresponding to primary, secondary, associative, and limbic cortices. Corresponding numerical values of cortical hierarchy levels were set to be from 1 to 4, respectively. To compare lipid intensity data to rs-fMRI data, a functional connectivity (FC) matrix was constructed for a subset of 59 brain regions, as described in the corresponding section of Methods. Outer product matrices were built for each lipid, and the corresponding correlation coefficient with the FC matrix was calculated.

### Reporting summary

Further information on research design is available in the Nature Portfolio Reporting Summary linked to this article.

## Data availability

The raw and processed RNA-seq data generated in this study for two brain regions have been deposited in the GEO database under accession number GSE262948. The RNA-data data used in this study for the remaining 33 brain regions are available under accession number GSE127898. All other data generated in this study are provided in the Supplementary Information and Source Data file. Source data are provided with this paper.

## Code availability

All custom code used in this manuscript is publicly available at https://doi.org/10.5281/zenodo.10908108.

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

## Acknowledgements

The authors are grateful to the CoBrain IT team and personally to Dmitry Vinogradov for the technical support. The reported study was funded by the Russian Science Foundation under grant number 22-15-00474 (P.K., O.E., E.S., M.O., A.T., N.A.). M.O.'s work was additionally funded by RFBR according to the research project № 20-34-90146. RNA-seq analysis was supported by the Russian Science Foundation (grant number 21-74-10102 to E.K.). The project was partially supported by the Moscow State University Program of Development (FACSAria SORP cell sorter) (D.P.). Work in the M.W. laboratory is supported by grants from the National University of Singapore via the Life Sciences Institute (LSI), the National Research Foundation (NRFSBP-P4), and the NRF and A*STAR IAF-ICP I1901E0040. ToF-SIMS measurements were performed in the FRCCP RAS Center of the Collective Equipment (no. 506694) (A.G., A.V.).

## Author contributions

P.K. designed the research, W.M., J.C.F, N.A., P.G., O.E., G.V., D.P., A.G., A.V., and E.S. performed the experiments, E.E.K., M.O., A.T., A.M., M.R.W., P.O., A.S., S.G., M.F., E.N., Y.K., and I.K. analyzed the data. All authors contributed to writing the paper.

## Competing interests

The authors declare no competing interests.
