## [Peer Review File · Nature Communications]

Lipidome atlas of the adult human brainREVIEWER COMMENTS

Reviewer #1 (Remarks to the Author):

Comments for the authors.

The work entitled 'Comprehensive map of the human brain lipidome' presents data related to the role of lipids in brain using a lipidomic approach applied to humans but extended also to different experimental models (primates and rodents). The number of regions of the human brain analyzed is especially commendable, which makes the present study unique.

The paper is a correctly written and carefully prepared manuscript, resulting in a very attractive contribution to the field. The lipidomic approach, data analysis, preparation and presentation of results (figures and tables) -considering the extraordinary volume of data-, the complementary methods and analytical approaches used (transcriptomics, brain imaging, generation of original lipid 'functional' categories, etc.), and extension of findings to additional animal species, among other considerations, is excellent. I would also like to highlight the excellent identification of the limitations of the study, which, however, does not detract one iota of value from the results presented in this invaluable and innovative study. However, although the premise of the work is very stimulating, some minor concerns need to be addressed as suggestions for manuscript improvement.

Minor concerns:

In my opinion, the introduction and discussion sections need a major elaboration. I'm aware of the editorial limitations in manuscript extension and references' number. However, some additional information must be added even briefly.

Introduction. Previous works analyzing lipid classes/subclasses and fatty acid compositional profiles in both whole brain (gray matter and white matter) and different regions of the human brain should be recognized. See for example:

Sastry, P.S. Lipids of Nervous Tissue: Composition and Metabolism. *Prog. Lipid Res.* 1985, 24, 69–176, doi:10.1016/0163-7827(85)90011-6.

Thudichum, J.L.. *A Treatise on the Chemical Constitution of the Brain*; Hamden: Archon Books: Connecticut, USA, 1962.

Naudí, A.; Cabré, R.; Ayala, V.; Jové, M.; Mota-Martorell, N.; Portero-Otín, M.; Pamplona, R. Region-Specific Vulnerability to Lipid Peroxidation and Evidence of Neuronal Mechanisms for Polyunsaturated Fatty Acid Biosynthesis in the Healthy Adult Human Central Nervous System. *Biochim. Biophys. Acta - Mol. Cell Biol. Lipids* 2017, 1862, doi:10.1016/j.bbalip.2017.02.001.

Mota-Martorell, N.; Andrés-Benito, P.; Martín-Gari, M.; Galo-Licona, J.D.; Sol, J.; Fernández-Bernal, A.; Portero-Otín, M.; Ferrer, I.; Jove, M.; Pamplona, R. Selective Brain Regional Changes in Lipid Profile with Human Aging. *GeroScience* 2022, 44, 763–783, doi:10.1007/S11357-022-00527-1.

Rouser, G.; Galli, C.; Kritchevsky, G. Lipid Class Composition of Normal Human Brain and Variations in Metachromatic Leucodystrophy, Tay-Sachs, Niemann-Pick, Chronic Gaucher's and Alzheimer's Diseases. *J. Am. Oil Chem. Soc.* 1965, 42, 404–410, doi:10.1007/BF02635576.

O'Brien, J.S.; Sampson, E.L.; Brien, O.; Fillerup, D.L.; Mead, J.F.; Lz, J. Lipid Composition of the Normal Human Brain: Gray Matter, White Matter, and Myelin". 1965, 5, 329, doi:10.1016/S0022-2275(20)39619-X.

Söderberg, M.; Edlund, C.; Kristensson, K.; Dallner, G. Lipid Compositions of Different Regions of the Human Brain during Aging. *J. Neurochem.* 1990, 54, 415–423.

Svennerholm, L. Distribution and Fatty Acid Composition of Phosphoglycerides in Normal Human Brain. 1968, doi:10.1016/S0022-2275(20)42702-6.

Discussion. In my opinion, the discussion is excessively poor and unspecific. Thus:

The functional significance of the prevalence of different lipid classes is not evaluated. The functional significance of the cross-regional prevalence of different lipid classes is not assessed. Are there lipidomic differences between cortical regions/areas? And between cortical and subcortical regions? Could any functional significance be attributed? Could differences between regions have evolutionary significance? Are the results obtained at the level of gray and white matter (myelin+ and myelin-) aligned with previous studies? Have the results obtained in relation to the degree of unsaturation been contrasted with previous studies? Are they comparable? Can we attribute some functional significance to the lipidomic differences found according to cell type? I am aware that some questions do not currently have an answer, given our limited knowledge of the biology of lipids in the human brain. However, the authors should make an additional effort to interpret, even partially and briefly, the results obtained.

Reviewer #2 (Remarks to the Author):

In their manuscript Osetrova and colleagues present a lipidomic atlas of the human brain. Their motivation is to bridge the gap between 'macro' data such as functional MRI and 'micro' data such as bulk tissue RNA-seq data. Using mass spectrometry, they describe the lipid composition (lipidome) across 75 brain regions and attempt to relate the unique composition of these regions to their function (functional connectivity). The Authors had previously generated RNA-seq data from these regions (PMID: 32424074; augmented further in this study) and show greater correlations between the lipidome and functional correlates than the brain transcriptome.

The techniques employed here and overlapping nature of techniques such as HRMS/MRM and MALDI-Imaging have resulted in a high-quality data set. However, it is less clear whether, armed with this rich data, the Authors have used it effectively to further knowledge in this area. The Authors have rather taken a conservative stance with parcellation of their data into just five classes: myelin+ and myelin-, unexplained and housekeeping and variable lipids. This creates a more confirmatory tone towards existing data rather than being 'ground-breaking'. The Authors should consider a more nuanced classification.

The linking of lipid profiles to cell types e.g. astrocytes and unexplained lipids was more interesting but an opportunity to then relate these profiles to that cell's known function(s) was missed in favour of the so-called hierarchy of signalling processing (HR). This latter 'macro' approach (relating lipidomes to a 4 level-cortical hierarchy) seemed too general or even too ambitious to be useful here. More specifically, it seems paradoxical that HR- lipids were overrepresented in the myelin- category (line 194) and then that HR- lipids were associated with oligodendrocytes (line 204). In contrast the associations with the rs-fMRI data were more predictable, and informative.

Overall, the current manuscript needs to do a better job in the Introduction of informing the reader of the current state of human brain lipidomics and where the knowledge gaps exist. The Authors should then set up some hypotheses around these knowledge gaps e.g. cell lipid composition versus known cell functions to illustrate the utility of the lipidome. Lipid classes should also be briefly introduced and the scope (limits) of the techniques used here for differentiating between subclasses. The data generated here is extensive, but the classification employed is too basic. The data deserves a more granular exploration of lipid sub-classes and their associations to cell type (even subtype) and cell function. The current associations with brain function, particularly HR, seem too ambitious here, and the Authors might consider omitting in favour of fMRI data only. A major omission in the Discussion is the Authors' opinions of how this data can be best applied to answer key questions about brain physiology and pathology. For example, diseases affecting oligodendrocytes versus neurons (neuronal sub-types) could be used as exemplars for 'painting a picture' where lipidome (cell specificities) or anomalies warrant investigation. This would reinforce the contention that the lipidome is "bridging the

two research streams" (line 49).

Minor points

Line 31 – 56 cortical regions in Results (line 77)

Line 36 - associations

Line 39 - the targeted....

Line 44 – represents the most....

Line 56 'formatting'

Line 57 – have uncovered

Line 62 – make the distinction between structures and regions clearer here

Line 72 – include gender and age of the donors.

Line 89 – lipid classes and potentially comment on overlap (n = 169) between the two techniques – is this lower than expected?

Line 119 - lipids in the gray matter

Line 161 – myelin is oligodendrocyte processes

Line 169 – revealed similarities

Line 187 – State the four levels, also state the Authors rather than just citation.

Line 250 – contain an excess

Line 387 – agonal

Line 388 – was their ethics approval from a local committee for this study

Line 395 – were kept at -20C....

Line 403 – 10-15 mg seems low, please confirm.

Line 410 – Only limited number of internal standards available; please comment

Line 420 – 14,000 g

Line 527 – standards

Line 648 – Include Authors, not just citation

Line 785 – Include the matrix used and whether any calibrants were included

Line 851 - built

Reviewer #3 (Remarks to the Author):

The study conducted involved a comprehensive lipidomic mapping of human and macaque brains. The study is interesting from the standpoint of elucidating which lipid species are present in the various regions studies. The authors also utilized targeted methods to validate HRMS findings and MRI to derive structural features.

A major issue with the paper is that some of the molecular characterizations do not make sense, which points to the problem of using untargeted HRMS analysis for this type of work. For instance, adrenic acid enrichment is surprising given that omega-6 arachidonic acid is quantitatively much more abundant in the human brain than omega-6 adrenic acid. In Figure 6, linoleic acid seems to be enriched in myelin (almost as much as 18:1-9), but it is thousands of times less abundant than 18:1-9. In fact, the human brain has very little linoleic acid containing 2 double bonds. 10 double bonds in a molecule? This is a bit unusual even if the mass-spec says so.

HRMS does not correct for the response factor of each molecule making relative quantitation meaningless. To exemplify, in Figure 3- the authors show that cholesterol myelin content is low, when in fact, cholesterol is enriched in myelin and various brain regions (free cholesterol makes up 20% of the brain total lipid content).

The mRNA-lipidomic overlay is confusing...

In the methods, sample sizes need to be provided upfront.

The authors describe a mouse experiment in the method which is not discussed or presented...?

The authors state that the brain is 50% lipids without providing a reference. I think a more recent paper disputes that figure: PMID: 36196762

The authors state that macaques were sacrificed for reasons other than participation on the study – further clarification would be useful.

When were the MRI images taken relative to the time of death (For humans and macaques)?

Response to Reviewer Comments

Reviewer #1 (Remarks to the Author):

Comments for the authors.

The work entitled ‘Comprehensive map of the human brain lipidome’ presents data related to the role of lipids in brain using a lipidomic approach applied to humans but extended also to different experimental models (primates and rodents). The number of regions of the human brain analyzed is especially commendable, which makes the present study unique.

The paper is a correctly written and carefully prepared manuscript, resulting in a very attractive contribution to the field. The lipidomic approach, data analysis, preparation and presentation of results (figures and tables) -considering the extraordinary volume of data-, the complementary methods and analytical approaches used (transcriptomics, brain imaging, generation of original lipid ‘functional’ categories, etc.), and extension of findings to additional animal species, among other considerations, is excellent. I would also like to highlight the excellent identification of the limitations of the study, which, however, does not detract one iota of value from the results presented in this invaluable and innovative study. However, although the premise of the work is very stimulating, some minor concerns need to be addressed as suggestions for manuscript improvement.

Minor concerns:

1. In my opinion, the introduction and discussion sections need a major elaboration. I’m aware of the editorial limitations in manuscript extension and references’ number. However, some additional information must be added even briefly.

Introduction. Previous works analyzing lipid classes/subclasses and fatty acid compositional profiles in both whole brain (gray matter and white matter) and different regions of the human brain should be recognized. See for example:

Sastry, P.S. Lipids of Nervous Tissue: Composition and Metabolism. *Prog. Lipid Res.* 1985, 24, 69–176, doi:10.1016/0163-7827(85)90011-6.

Thudichum, J.L.. *A Treatise on the Chemical Constitution of the Brain*; Hamden: Archon Books: Connecticut, USA, 1962.

Naudí, A.; Cabré, R.; Ayala, V.; Jové, M.; Mota-Martorell, N.; Portero-Otín, M.; Pamplona, R. Region-Specific Vulnerability to Lipid Peroxidation and Evidence of Neuronal Mechanisms for Polyunsaturated Fatty Acid Biosynthesis in the Healthy Adult Human Central Nervous System. *Biochim. Biophys. Acta - Mol. Cell Biol. Lipids* 2017, 1862, doi:10.1016/j.bbalip.2017.02.001.

Mota-Martorell, N.; Andrés-Benito, P.; Martín-Gari, M.; Galo-Licon, J.D.; Sol, J.; Fernández-Bernal, A.; Portero-Otín, M.; Ferrer, I.; Jove, M.; Pamplona, R. Selective Brain Regional Changes in Lipid Profile with Human Aging. *GeroScience* 2022, 44, 763–783, doi:10.1007/S11357-022-00527-1.

Rouser, G.; Galli, C.; Kritchevsky, G. Lipid Class Composition of Normal Human Brain and Variations in Metachromatic Leucodystrophy, Tay-Sachs, Niemann-Pick, Chronic Gaucher’s and Alzheimer’s Diseases. *J. Am. Oil Chem. Soc.* 1965, 42, 404–410, doi:10.1007/BF02635576. O’Brien, J.S.; Sampson, E.L.; Brien, O.’; Fillerup, D.L.; Mead, J.F.; Lz, J. Lipid Composition of

the Normal Human Brain: Gray Matter, White Matter, and Myelin". 1965, 5, 329, doi:10.1016/S0022-2275(20)39619-X.

Söderberg, M.; Edlund, C.; Kristensson, K.; Dallner, G. Lipid Compositions of Different Regions of the Human Brain during Aging. *J. Neurochem.* 1990, 54, 415–423.

Svennerholm, L. Distribution and Fatty Acid Composition of Phosphoglycerides in Normal Human Brain. 1968, doi:10.1016/S0022-2275(20)42702-6.

We appreciate the Referee's comment. In line with the Referee's suggestion, we have significantly revised and expanded the Introduction section of our manuscript. This includes the incorporation of suggested references and several other recent studies.

Discussion. In my opinion, the discussion is excessively poor and unspecific. Thus:
2. The functional significance of the prevalence of different lipid classes is not evaluated.

*We are grateful for the Referee's comment. In response, we have thoroughly revised the Discussion section of our manuscript. Among other modifications, we have included a new paragraph that delves into the known and potential functional significance of specific lipid class abundance distribution, particularly in relation to the variation observed across the brain regions under investigation. Moreover, we have incorporated an additional analysis focusing on the prevalence of lipid class and fatty acid residues across the human brain. The findings of this analysis are presented in the newly added **Figure 2**. In addition, we have made alterations to **Figure 4** (former **Figure 3**) to more accurately depict the biochemical properties of lipids prevalent in each of the five categories that were defined based on the lipid abundance distribution across the brain. These changes are now reflected both in the Results and Discussion sections of the manuscript.*

3. The functional significance of the cross-regional prevalence of different lipid classes is not assessed.

We appreciate the Referee's comment. We have now significantly modified the Discussion section to address this issue. In particular, we included discussion of our results showing the relationship between lipids and lipid classes with regional myelin content obtained from sMRI measurements, major cell type composition evaluated using marker gene expression profiles, processing hierarchy based on anatomical and electrophysiological data, and functional connectivity ascertained from resting state fMRI recordings. Moreover, we compared the observed variations in lipid class prevalence with the lipid composition of brain regions, cell types, and myelin membranes reported in existing literature. This information has been integrated into the revised Discussion section of our manuscript (lines 256-320 in the revised version).

4. Are there lipidomic differences between cortical regions/areas? And between cortical and subcortical regions? Could any functional significance be attributed?

We appreciate the Referee's feedback. Indeed, our results show the existence of lipid abundance variations between cortical areas, as well as between the neocortex and subcortical structures. To assign functional significance to these differences, in the absence of well-established functional annotation of individual lipid compounds, we explored the functional implications of variations in lipid class prevalence across different brain regions, including differences among neocortical regions and between neocortical regions and other brain structures, by associating them with

variation in regional myelin content obtained from sMRI measurements, major cell type composition evaluated using marker gene expression profiles, processing hierarchy based on anatomical and electrophysiological data, and functional connectivity ascertained from resting state fMRI recordings. We acknowledge that this method of functional annotation is indirect and does not match the level of detail available for individual proteins or their mRNA transcripts. Our findings, however, provide a comprehensive overview of the relationship between lipid abundance variation across brain regions and the underlying structural and functional brain properties. To better communicate this point, we have revised the Discussion section of our manuscript (lines 283-320 in the revised version).

5. Could differences between regions have evolutionary significance?

We are grateful to the Referee for this comment. Our dataset indeed extends beyond human lipidome data, additionally encompassing lipidome measurements from adult rhesus monkey brains. Our comparative analysis of lipid abundance variation profiles between human and macaque brains revealed a strong conservation of variation patterns across all lipids. The exception to this were lipids that displayed stable variation profiles, which could not be explained by variations in myelin content (unexplained lipid category). Interestingly, we found these very same lipids to have the strongest association with the signal processing hierarchy architecture of the human brain. This suggests that this lipid group may play a role in shaping the properties of information processing networks. We have incorporated this information into the revised Discussion section of our manuscript (lines 287-293 in the revised version).

6. Are the results obtained at the level of gray and white matter (myelin+ and myelin-) aligned with previous studies?

We are grateful to the Referee for bringing up this issue. In our previous version of the manuscript we briefly described the comparison between lipids designated into myelin⁺ and myelin⁻ categories and a study reporting lipid composition differences between cortical myelin-poor gray and myelin-rich white matter (lines 142-144 and 162-172 in the revised version) and reported agreement with previous publications in terms of lipid class representation between myelin⁺ lipids and lipid classes prevalent in myelin membranes (**Figure 4A**, former **Figure 3**). In our previous version we also included MALDI and SIMS-TOF based mass spectrometry imaging experiments placing myelin⁺ lipid signal within white matter structures and myelin⁻ lipid signal – within gray matter (lines 144-147 and 173-176 in the revised version).

In response to the Referee's suggestion, we have incorporated an analysis of lipid class and fatty acid residue unsaturation distribution among various brain regions, as depicted in the new **Figure 2**. The results clearly align with reported differences between myelin-rich and myelin-poor brain regions in terms of lipid class and fatty acid unsaturation extent. Importantly, we found that lipid classes previously identified as enriched in myelin membranes, such as cholesterol, DGs, and sphingolipids (specifically ceramides, HexCers, and SulfoHexCers), are overrepresented in myelin-rich brain regions, representing white matter of the brain. We have now included these findings and discussed the consistency between previous studies and our results throughout the manuscript in the Discussion section (lines 256-266 in the revised version).

7. Have the results obtained in relation to the degree of unsaturation been contrasted with previous studies? Are they comparable?

*We are grateful to the Referee for bringing up this issue. Following the Referee's suggestion, we have added analysis of lipids' fatty acid residue unsaturation distribution among brain regions illustrated in new **Figure 2**. The results agree with reported fatty acid unsaturation extent differences between myelin-rich and myelin-poor brain regions. Specifically, we show that in agreement with previous reports (study "Cell-type- Brain-Region-Resolved Mouse Brain Lipidome", study "A metabolome atlas of the aging mouse brain"), lipidome composition of myelin-rich regions, such as subcortical white matter tracks, contain greater proportion of saturated and oligo-unsaturated fatty acid residues compared to myelin-poor regions, such as neocortical gray matter and, especially, its prefrontal cortex. Further, results shown in **Figure 4A** (former **Figure 3**) demonstrate significant enrichment of PEs containing highly polyunsaturated fatty acids among myelin⁻ lipids negatively associated with myelin content and characteristic of neurons. These results similarly align with reported prevalence of polyunsaturated fatty acid residues in PC and PE lipids comprising the major proportion of plasma membrane lipidome of neurons and astrocytes prevalent in the myelin-poor brain structures (study "Cell-type- Brain-Region-Resolved Mouse Brain Lipidome", review "Polyunsaturated fatty acids and their metabolites in brain function and disease"). We now added these results and described the agreement between previous studies and results presented throughout the manuscript to the Discussion section (lines 119-127 and 265-266 in the revised version).*

8. Can we attribute some functional significance to the lipidomic differences found according to cell type?

We are grateful to the Referee for this comment. Given the current state of lipid functional annotation, which lags behind that of proteins, we sought to assign function to lipids by associating them with primary brain cell types. Prior studies have already highlighted differences in membrane lipid composition among brain cells, with myelin membranes exhibiting the most distinct composition. Our findings not only corroborate these previous results but also provide an additional assignment of individual lipids to inhibitory and excitatory neurons, astrocytes, microglia, and oligodendrocyte progenitor cells. This annotation, although indirect, was supported by comparison to lipid composition of isolated mouse neurons and published mouse brain cell lines included in our study. Such annotation, therefore, could serve as a foundation for future studies of the human brain. For example, identified lipid composition differences between disease and control samples could be attributed to specific cell types. We have now incorporated this point into the revised manuscript (lines 283-293 in the revised version).

I am aware that some questions do not currently have an answer, given our limited knowledge of the biology of lipids in the human brain. However, the authors should make an additional effort to interpret, even partially and briefly, the results obtained.

Reviewer #2 (Remarks to the Author):

In their manuscript Osetrova and colleagues present a lipidomic atlas of the human brain. Their motivation is to bridge the gap between ‘macro’ data such as functional MRI and ‘micro’ data such as bulk tissue RNA-seq data. Using mass spectrometry, they describe the lipid composition (lipidome) across 75 brain regions and attempt to relate the unique composition of these regions to their function (functional connectivity). The Authors had previously generated RNA-seq data from these regions (PMID: 32424074; augmented further in this study) and show greater correlations between the lipidome and functional correlates than the brain transcriptome.

The techniques employed here and overlapping nature of techniques such as HRMS/MRM and MALDI-Imaging have resulted in a high-quality data set. However, it is less clear whether, armed with this rich data, the Authors have used it effectively to further knowledge in this area. The Authors have rather taken a conservative stance with parcellation of their data into just five classes: myelin+ and myelin-, unexplained and housekeeping and variable lipids. This creates a more confirmatory tone towards existing data rather than being ‘ground-breaking’.

1. The Authors should consider a more nuanced classification.

We thank the Referee for this comment. Our categorization of lipids into five groups, based on their abundance profiles across brain regions, is indeed a simplified representation of the complex variations in lipid abundance within the human brain. Prior research has underscored the distinct composition of myelin lipids, which significantly deviates from that of other cell plasma membranes, as a key driver of lipid abundance variation among brain regions. Our findings are consistent with this view, but also unveil lipid variation profiles that are not dictated by myelin, spanning three out of the five categories we proposed. Furthermore, we have conducted a comprehensive characterization of the lipids within these five categories using a multi-tiered classification system. This system takes into account various factors including biochemical lipid classes, cell type assignment, and association with functional brain features such as connectivity. In response to the Referee's comment, we have made revisions to the manuscript, particularly in the Discussion sections, to underscore the diverse lipid properties examined in our study (lines 256-312 in the revised version).

*Additionally, to provide a more nuanced view of lipid abundance variation among the investigated brain regions, we have included an analysis at the individual lipid class level, encompassing a total of 21 classes shown in new **Figure 2**. This analysis further illustrates the complexity of lipid distribution and its variation across different brain regions, with myelin-associated lipid classes showing the largest variation among brain regions (lines 119-127 in the revised version).*

2. The linking of lipid profiles to cell types e.g. astrocytes and unexplained lipids was more interesting but an opportunity to then relate these profiles to that cell's known function(s) was missed in favour of the so-called hierarchy of signalling processing (HR). This latter ‘macro’ approach (relating lipidomes to a 4 level-cortical hierarchy) seemed too general or even too ambitious to be useful here.

We appreciate the Referee's feedback. In response to your suggestion, we have amended the Discussion section of our manuscript to better connect our findings with the known functions of different cell types. We would also like to highlight that our analysis of lipid abundance association with signal processing hierarchy, while it may seem speculative, is grounded in recent research.

Specifically, the study "Single-cell spatial transcriptome reveals cell-type organization in the macaque cortex" demonstrated a correlation between gene expression levels and the hierarchy of visual signal processing within the macaque cortex. Consequently, we believe that although our hierarchy analysis certainly warrants further investigation, its inclusion in our manuscript could provide useful insights for future human and animal brain lipidome studies. To ensure clarity on this point, we have revised the manuscript accordingly (lines 306-310 in the revised version).

Moreover, we supplemented our analysis with correlation of second principal component of lipid intensities matrix with hierarchy level for individual processing pathways (auditory, somatosensory, and visual) (see **Figure R1** below).

In line with results on macaque visual system and transcriptome, PC2 showed significant positive correlation to processing hierarchy levels in visual network. Similarly, among individual lipids myelin- category demonstrated most significant association, while myelin+ lipids had greater amplitude of linear model slope.

3. More specifically, it seems paradoxical that HR- lipids were overrepresented in the myelin- category (line 194) and then that HR- lipids were associated with oligodendrocytes (line 204). In contrast the associations with the rs-fMRI data were more predictable, and informative.

We are grateful to the Referee for highlighting this confusing point. We have shown that at the category level, myelin-associated lipids exhibit the most significant negative correlations with processing hierarchy, as evidenced by the Mann-Whitney test. While oligodendrocyte-associated lipids predominantly show a negative correlation with processing hierarchy, consistent with greater reported myelination of primary sensory regions, and larger differences in terms of effect amplitude. This amplitude effect, however, is not reflected in the p-value due to high intragroup variability. This result creates an apparent discrepancy between panels C and G of the figure, as panel C primarily reflects the correlation significance for the entire category, while panel G focuses on the association effect strength for a limited number of individual lipids that pass the significance threshold. We have revised the figure legend to better explain this point (lines 439-442 in the revised version).

4. Overall, the current manuscript needs to do a better job in the Introduction of informing the reader of the current state of human brain lipidomics and where the knowledge gaps exist.

We appreciate the Referee's feedback on this matter. In response to the suggestion, we have extensively revised and expanded the Introduction section of our manuscript. This includes the integration of the original studies characterizing human brain lipids, as well as more recent work, highlighting remaining gaps of knowledge, as suggested (lines 51-76 in the revised version).

5. The Authors should then set up some hypotheses around these knowledge gaps e.g. cell lipid composition versus known cell functions to illustrate the utility of the lipidome.

We thank the Referee for this suggestion. In our study we are indirectly assessing the involvement of lipids in shaping structural and functional properties of the human brain by testing the relationship between the lipidome composition and these properties. These properties include, for structural properties, cell type composition and myelin proportion of selected brain regions, their spatial proximity and shared anatomical origin; and for functional properties, signal processing hierarchy, functional connectivity, and evolutionary conservation to macaques. By examining the correlations between the lipidome composition and these properties, our study aims to provide novel insights into the relationship between lipidome composition and brain organization and function. We have revised our manuscript to include this hypothesis (lines 51-76 and 252-257 in the revised version).

6. Lipid classes should also be briefly introduced and the scope (limits) of the techniques used here for differentiating between subclasses. The data generated here is extensive, but the classification employed is too basic.

We thank the Referee for bringing up this issue. We have provided a description of the limitations of used HPLC-MS techniques within the "Lipid Identification" section of our manuscript. Specifically, we mentioned that the employed HLPC-MS method did not allow us to differentiate between certain types of isomers. The relevant part of the manuscript now states: "It implies that sn-attachments of FA, positions of double bonds, and stereochemistry are not declared" (lines 615-616 in the revised version). Our entire analysis and lipid classification in the current study are based on this premise, which acknowledges these limitations.

7. The data deserves a more granular exploration of lipid sub-classes and their associations to cell type (even subtype) and cell function.

We are grateful to the Referee for this comment. We concur that a comprehensive lipid annotation, in relation to morphologically and functionally defined cell types and possibly different histological structures in neuronal and glial cell projections, would substantially enhance our understanding of lipid roles and functions in the human brain. However, such detailed annotation of individual lipid compounds is currently in its early stages, with only a few studies offering results in this direction using isolated neurons or cultured mouse brain cells (study "Cell-type-Brain-Region-Resolved Mouse Brain Lipidome").

In our study, we were constrained by the indirect assignment of lipid profiles to cell types, which was based on the deconvolution of tissue sample composition using known cell type mRNA markers. Although this method is well established, it does have its limitations. In particular, while this deconvolution procedure is effective at separating major cell types, it struggles with cellular subtypes, which often display strongly correlated expression profiles. The same applies to lipids: while different lipid classes tend to show distinct abundance profiles across brain regions, lipid subclasses are often closely correlated.

In conclusion, we believe that the suggested analysis would greatly benefit from direct measurements of lipid composition of isolated cell types, subtypes, and histological structures. While this suggestion seems straightforward, its implementation, especially in the case of the human brain, requires significant effort, particularly in terms of proper sample procurement. Therefore, while we are certainly aiming to pursue this research direction, such work lies beyond the scope of the current study. Nevertheless, we hope that our current work might stimulate interest from other research groups in this study area. We have incorporated more discussion on potential functions of different lipid classes in the Discussion sections and have attempted to further underscore the need for greater effort towards functional lipid compound annotation (lines 283-293 and 332-336 in the revised version).

8. The current associations with brain function, particularly HR, seem too ambitious here, and the Authors might consider omitting in favour of fMRI data only.

We appreciate the Referee's comment. In this study, we aimed to provide a multifaceted interpretation of the human brain lipidome composition, hence our decision to include hierarchy analysis. Although the hierarchical organization of signal processing in the brain is often viewed from a functional perspective, it must be fundamentally based on structural and cytoarchitectural foundations. While certain aspects of these structural determinants, such as interregional neocortical connectivity architecture, have been explored, the molecular determinants of the hierarchy remain less studied.

As we mentioned in response to one of the previous comments, our analysis of lipid abundance association with signal processing hierarchy parallels recent gene expression study in conjunction with the visual cortex signal processing hierarchy (study "Single-cell spatial transcriptome reveals cell-type organization in the macaque cortex"). We, therefore, believe that our analysis contributes insights for future research into the human and animal brain lipidome. To ensure clarity on this point, we have revised the manuscript accordingly (lines 304-309 in the revised version).

9. A major omission in the Discussion is the Authors' opinions of how this data can be best applied to answer key questions about brain physiology and pathology. For example, diseases affecting oligodendrocytes versus neurons (neuronal sub-types) could be used as exemplars for 'painting a picture' where lipidome (cell specificities) or anomalies warrant investigation. This would reinforce the contention that the lipidome is "bridging the two research streams" (line 49).

We are grateful to the Referee for pointing out this omission. In response to this suggestion, we have revised and expanded the Discussion section of our manuscript. We now emphasize potential associations between our findings and previously reported alterations in the brain lipidome

associated with psychiatric and cognitive disorders across brain regions. This addition is aimed at providing a more comprehensive interpretation of our results in the broader context of neuropathology. The relevant changes can be found on lines 346-350 in the revised version of our manuscript.

10. Minor points

Line 31 – 56 cortical regions in Results (line 77)

Line 36 – associations

Line 39 - the targeted....

Line 44 – represents the most....

Line 56 ‘formatting’

Line 57 – have uncovered

Line 62 – make the distinction between structures and regions clearer here

Line 72 – include gender and age of the donors.

Line 89 – lipid classes and potentially comment on overlap (n = 169) between the two techniques
– is this lower than expected?

Line 119 - lipids in the gray matter

Line 161 – myelin is oligodendrocyte processes

Line 169 – revealed similarities

Line 187 – State the four levels, also state the Authors rather than just citation.

Line 250 – contain an excess

Line 387 – agonal

Line 388 – was their ethics approval from a local committee for this study

Line 395 – were kept at -20C.....

Line 403 – 10-15 mg seems low, please confirm.

Line 410 – Only limited number of internal standards available; please comment

Line 420 – 14,000 g

Line 527 – standards

Line 648 – Include Authors, not just citation

Line 785 – Include the matrix used and whether any calibrants were included

Line 851 – built

We thank the Referee for pointing out these errors. We have corrected them in the revised manuscript.

Reviewer #3 (Remarks to the Author):

The study conducted involved a comprehensive lipidomic mapping of human and macaque brains. The study is interesting from the standpoint of elucidating which lipid species are present in the various regions studies. The authors also utilized targeted methods to validate HRMS findings and MRI to derive structural features.

1. A major issue with the paper is that some of the molecular characterizations do not make sense, which points to the problem of using untargeted HRMS analysis for this type of work. For instance, adrenic acid enrichment is surprising given that omega-6 arachidonic acid is quantitatively much more abundant in the human brain than omega-6 adrenic acid.

*We thank the Referee for drawing attention to the result showing an enrichment of adrenic acid among lipids with a positive association with processing hierarchy levels. We understand that the Referee may have expected results on arachidonic acid, but due to the isomeric nature of omega-3 eicosapentaenoic acid, we cannot deduce this from the brutto formula. We acknowledge the limitations of our method and intend to investigate the fatty acid composition of the brain more thoroughly in future research. Nevertheless, we have replicated the analysis presented and found similar enrichment for total 20:4 FA, which is likely dominated by arachidonic acid, a component known to be abundant in the human brain (see **Figure R2** below).*

2. In Figure 6, linoleic acid seems to be enriched in myelin (almost as much as 18:1-9), but it is thousands of times less abundant than 18:1-9. In fact, the human brain has very little linoleic acid containing 2 double bonds.

*We thank the Referee for drawing attention to the previously confusing results presented in **Figure 6** (now **Figure 7**). In the previous version, we plotted the number of lipids associated with the category rather than their intensity, and we considered the total number of double bonds per lipid, not the number of double bonds in a single fatty acid. We acknowledge that lack of individual fatty acid residue unsaturation level resolution may limit the interpretation of the results to some extent.*

We believe, however, that these data still provide an approximation of individual fatty acid unsaturation content, given the typical placement of saturated or polyunsaturated residues at the first position. To clarify this, we have updated the figure and its description. The distribution is now divided into several sub-distributions, each representing a group of lipids with the same number of carbon chains.

3. 10 double bonds in a molecule? This is a bit unusual even if the mass-spec says so.

Following up on the response to the previous comment, we would like to clarify that we originally considered the total number of double bonds per molecule, not the number of double bonds in each fatty acid for the results representation. We understand the confusion this may have caused and have accordingly modified the figure and its legend as described above (lines 457-459 in the revised version).

4. HRMS does not correct for the response factor of each molecule making relative quantitation meaningless. To exemplify, in Figure 3- the authors show that cholesterol myelin content is low, when in fact, cholesterol is enriched in myelin and various brain regions (free cholesterol makes up 20% of the brain total lipid content).

*We are grateful to the Referee for bringing attention to this ambiguous result. In former **Figure 3** (now **Figure 4**) we illustrate the number of identified lipid compounds per lipid class, rather than their abundance. Consequently, cholesterol is represented by a single compound, leading to a small peak. To better depict the relative abundances of evaluated lipid classes, we have included an additional analysis shown in new **Figure 2**. This figure presents normalized intensities of lipid classes and lipids, sorted by the cumulative unsaturation of their fatty acid residues. As can be seen in **Figure 2**, cholesterol levels are indeed high in myelin-rich structures, which is consistent with prior observations. We have now revised our manuscript to further clarify this point (lines 119-127 and 403-416 in the revised version).*

5. The mRNA-lipidomic overlay is confusing...

We appreciate the Referee's feedback. We decided to extract this part of analysis from this manuscript and investigate this relationship in more detail in another manuscript.

6. In the methods, sample sizes need to be provided upfront.

We thank the Referee for pointing out this issue. We have now included sample size information in the Methods section wherever appropriate.

7. The authors describe a mouse experiment in the method which is not discussed or presented...?

Results on sorted cells from mouse brain lipidome were presented on Figure 4L-M in the former version of the manuscript and on Figure 5G-H in the modified version.

8. The authors state that the brain is 50% lipids without providing a reference. I think a more recent paper disputes that figure: PMID: 36196762

We are grateful to the Referee for highlighting this discrepancy. There are indeed varying percentages of lipid proportion in the brain as presented in different sources of literature. As the Referee has pointed out, there is a distinction between lipid proportions in what is referred to as "fresh tissue weight" versus "dry tissue weight". Taking the Referee's suggestion into account, we have conducted a thorough review of relevant literature and have now incorporated more accurate and referenced estimates of lipid contributions to brain tissue composition in both myelin-rich and myelin-poor tissues (please refer to lines 51-52 in the revised version).

9. The authors state that macaques were sacrificed for reasons other than participation on the study – further clarification would be useful.

We thank the Referee for this comment. The macaques used in our study were part of a control (untreated) group in a non-commercial study. This important detail has now been incorporated into the revised manuscript, in the sample description section (lines 477-480 in the revised version).

10. When were the MRI images taken relative to the time of death (For humans and macaques)?

We thank the Referee for this comment. For sMRI and fMRI data comparison with our own lipidome data, we utilized datasets from previously published research, as mentioned in the Methods section. Furthermore, we want to confirm that all MRI measurements in these published studies were obtained from living subjects, including both human and macaque individuals. We have updated our manuscript to clarify this point (lines 736-743 in the revised version).

REVIEWER COMMENTS

Reviewer #1 (Remarks to the Author):

The authors have made substantial revisions in their paper. The message of the paper and supporting illustrations are now clearer. I think they have addressed all my concerns.

Reviewer #2 (Remarks to the Author):

The Revised manuscript from Osetrova is greatly improved. The Authors have diligently attempted to meet previous queries while still acknowledging the limitations of lipidomics. One outstanding query is with regarding Fig 7, column 5 and the 'None' category of cell type. Could the Authors include a description of this category (as per the revised Methods (Line 793) in the legend? Then add a short sentences or two in the Discussion explaining the prominence of this 'non-category' and speculate on why this is occurring and how future analyses might derive more definitive cell types.

Reviewer #3 (Remarks to the Author):

The authors have addressed most comments. The paper, however, remains clouded with vagueness in some parts. Added clarity would help the reader understand the key take aways from the study.

Fis 2: what do the authors mean by 'lipid abundance'? Is this relative levels? I suggest using the term Relative levels or the like. Abundance is misleading as it indicates that the brain has a lot of TGs and DGs, which is not true. These amount to <1% of brain lipids. (same changes should be applied in text).

What do the authors mean exactly when they say lipid enrichment? Eg in Figure 4. Please define this explicitly (mathematically if possible) in both the methods and results section of the text.

Also define intensities. Is this peak area? Standard-normalized peak area?

For all figures please include the sample size per group. Readers shouldn't need to go back to the methods everytime to figure that out.

Line 164: The authors state: "The myelin- 164 lipids showed an overrepresentation of 165 two phospholipid classes, and housekeeping lipids — excess of free fatty acids". What do they mean by excess free fatty acids? There isn't much free fatty acids in the brain. >99% are esterified. In the same paragraph spell out the two phospholipid classes that were overrepresented in myelin- Please re-iterate what 'unexplained lipids' means in the results section.

The HR+ and HR- paragraph is not clear – what is HR+ , what is HR-?

In Results, the authors need to state upfront that the MRI Was from different human and macaque subjects (not the same ones used for the lipidomic analysis). This remains a major limitation of this study.

In the discussion, the authors state: "This category, termed housekeeping lipids, was 279 significantly enriched in short-chain unesterified fatty acids, aligning with their role as universal building blocks in brain lipid biosynthesis". This is simply not true (the review paper cited does not show that either). Also, which short chain unsaturated fatty acids are they referring to?

Response to Referee comments, round two.

REVIEWER

COMMENTS

Reviewer #1 (Remarks to the Author):

The authors have made substantial revisions in their paper. The message of the paper and supporting illustrations are now clearer. I think they have addressed all my concerns.

We are grateful to the Referee for a positive assessment of our work.

Reviewer #2 (Remarks to the Author):

The Revised manuscript from Osetrova is greatly improved. The Authors have diligently attempted to meet previous queries while still acknowledging the limitations of lipidomics. One outstanding query is with regarding Fig 7, column 5 and the 'None' category of cell type. Could the Authors include a description of this category (as per the revised Methods (Line 793) in the legend? Then add a short sentences or two in the Discussion explaining the prominence of this 'non-category' and speculate on why this is occurring and how future analyses might derive more definitive cell types.

We appreciate the Referee's positive evaluation of our work. Generally, lipids that are not assigned to any specific brain cell type comprise compounds that are ubiquitously present, as well as those where data intensity variation was either insufficient or not reproducible enough to correlate with known cell type marker variation levels. In response to the Referee's suggestions, we have now incorporated a description of the lipid category that is unassigned to any particular brain cell type in the legend of Figure 7. We have also added an explanation regarding the prominence of this category in the Discussion section of our manuscript (line 303 in the revised manuscript). It now reads:

To assess the functional significance of lipid level variation among brain regions, we utilized a gene expression-based tissue composition deconvolution procedure to assign lipids to the main brain cell types. This assignment process, which relied on the correlation between lipid intensity and mRNA expression profiles of cell type marker genes across 35 brain regions, proved particularly effective for myelin-dependent categories. A substantial proportion of lipids in the other three categories, particularly housekeeping and variable lipids that did not display consistent variation profiles across brain regions, remained, however, unassigned to specific cell types (Figure 5C and 7).

Reviewer #3 (Remarks to the Author):

The authors have addressed most comments. The paper, however, remains clouded with vagueness in some parts. Added clarity would help the reader understand the key take aways from the study.

Fig 2: what do the authors mean by 'lipid abundance'? Is this relative levels? I suggest using the term Relative levels or the like. Abundance is misleading as it indicates that the brain has a lot of TGs and DGs, which is not true. These amount to <1% of brain lipids. (same changes should be applied in text).

We are grateful to the Referee for pointing out this issue. Indeed, the data presented in Figure 2 does not depict the absolute abundance of lipids in the brain. Rather, as the Referee suggested, the data in Figure 2 represents the abundance levels of lipids from a given lipid class or residue unsaturation group in a specific brain region, relative to the rest of the brain. We acknowledge that our previous usage of terms such as "mean normalized signal intensity" and "relative abundance" interchangeably in the Figure 2 legend may have led to some confusion. To correct this, we have revised the Figure 2 legend to clearly define the term 'relative abundance' at the start of the legend, and have ensured its consistent use throughout the legend and the remainder of the manuscript, wherever appropriate. We further added an introduction of the data normalization procedure designed to minimize individual variation and, therefore, allow reproducible evaluation of the lipid abundance variation among brain regions in the human and macaque brains (line 108 of the revised manuscript). It reads:

To minimize the biological variation that is inevitably present among humans and, to a lesser extent, among macaque samples, we standardized average lipid abundance levels among individuals. This was achieved by dividing the abundance of each lipid in each of the 75 human or 38 macaque regions by its per-individual mean. Consequently, our subsequent analysis is based on these normalized lipid abundance levels. This approach allows us to estimate the relative representation of each lipid among the investigated brain regions. However, it reduces our ability to compare different lipids with each other in terms of their absolute abundance in the brain tissue.

In addition, we have also amended the title of Figure 2's legend to more accurately reflect its content. The revised Figure 2 legend now reads:

Figure 2. *Relative abundance profiles of lipids sorted by class and unsaturation across brain regions.*

A. *Heatmaps illustrating the relative abundance of each lipid class in every brain region. The relative abundance refers to the average signal intensity of each lipid class in a given brain region normalized to its global average intensity, calculated across all 35 regions, and then averaged across individuals. Each row represents a lipid class, each column corresponds to a brain region. The color bars underneath the plots here and in panel **B** indicate the anatomical assignment of brain regions as shown in **Figure 1D**. Lipid classes are grouped based on their geometry, which is illustrated in an insert on the right side of the figure. The gray bars on the right side represent the number of detected lipids in each lipid class (Table S3). The brain regions on the right side are colored according to the relative abundance levels of cholesterol. **B.** Heatmaps representing the relative abundance levels of lipids containing a defined total number of double bonds in their fatty acid residues (rows) across brain regions (columns), averaged across individuals. The gray bars on the right side represent the number of detected lipids in each fatty acid unsaturation group (Table S3). The brain regions on the*

right side are colored according to the relative abundance levels of lipids with a total of six double bonds in their fatty acid residues.

What do the authors mean exactly when they say lipid enrichment? Eg in Figure 4. Please define this explicitly (mathematically if possible) in both the methods and results section of the text.

We are grateful to the Referee for pointing out this issue. The statistical test used in enrichment analysis, namely hypergeometric test, was mentioned in Figure 4 description and briefly described the Methods section:

Lipid class, number of double bonds, and chain length enrichment of lipid categories were evaluated using a hypergeometric test, implemented in R as phyper() function.

Following the Referee's suggestion, we have included an explanation of the hypergeometric test principle in the Methods section of the manuscript (line 796 in the revised version of the manuscript). It now reads:

Lipid class, number of double bonds, and chain length enrichment of lipid categories were evaluated using a hypergeometric test, implemented in R as phyper() function. The hypergeometric test is employed to assess the statistical significance of drawing a specific number of successes from a population with a known quantity of successes. For instance, in the enrichment analysis of lipid classes, all measured lipids serve as the population. This population contains a specific number of lipids in a particular class, representing the possible number of successes. The lipids within a specific profile category represent the lipids we draw.

Also define intensities. Is this peak area? Standard-normalized peak area?

We thank the Referee for bringing up this point. We defined lipid intensities as peak areas further log₁₀-transformed and normalized to the sample mass and the standards. We have clarified this point in the Methods section of the manuscript (line 729 in the revised version of the manuscript). It now reads:

Lipid intensities, defined as areas under mass spectrometric peaks, were log₁₀-transformed, and all resulting lipid intensities were normalized on the median value of standards in a sample and the wet weight of the sample. For normalization wet weights and standard intensities were log₁₀-transformed, then the difference of sample values from the mean of a parameter was subtracted. Resulting lipid intensity values were further normalized using the mean intensity of this lipid calculated within each individual's brain to adjust for inter-individual variability.

For all figures please include the sample size per group. Readers shouldn't need to go back to the methods everytime to figure that out.

We thank the Referee for pointing out this issue. We have now added sample size numbers to figure legends, wherever appropriate.

Line 164: The authors state: "The myelin- 164 lipids showed an overrepresentation of 165 two phospholipid classes, and housekeeping lipids — excess of free fatty acids". What do they mean by excess free fatty acids? There isn't much free fatty acids in the brain. >99% are esterified.

We thank the Referee for highlighting this point. Indeed, as the Referee rightly points out, unesterified fatty acids only constitute a small fraction of all lipid species present in brain tissue. In our research, however, we focused on whether any specific lipids or lipid chemical classes, including minor ones, exhibit particular intensity variation patterns among brain regions. We feel that such approach is justified, as minor lipid classes could also have functional significance. Unesterified fatty acids, for example, have been shown to play crucial roles in brain signalling and metabolic processes. Our analysis reveals that unesterified fatty acids do indeed form a particular variation pattern among brain regions, an observation confirmed by a statistical (hypergeometric) test described in response to a previous question. Specifically, short chain fatty acids were found to be significantly associated with an intensity pattern that we classified as "housekeeping" in our study.

In the same paragraph spell out the two phospholipid classes that were overrepresented in myelin-

We thank the Referee for this comment. We have added two two phospholipid classes names, lysophosphatidylcholines and phosphatidylethanolamines, to the revised manuscript, as suggested (line 173 in the revised version of the manuscript).

Please re-iterate what 'unexplained lipids' means in the results section.

We are grateful to the Referee for this comment. The "unexplained" category description actually is first introduced in the Results section (line 160 in the revised manuscript). We have amended it for clarity. Now it reads:

"Among the remaining 18% of the analyzed lipidome, 11% (N = 46) differed among brain regions in a reproducible manner not explained by the myelin content (unexplained lipid category)."

Unexplained category is also explained ones again in the Discussion section of the manuscript (line 333 in the revised manuscript), which reads:

"It should be noted that the myelination signal does not determine the relationship between the composition of the lipidome and the signal processing hierarchy. This is evident from the fact that the unexplained lipid category, which is independent of myelin content, shows the strongest association with the brain's signal processing architecture (Figure 6C, Figure 7)."

The HR+ and HR- paragraph is not clear – what is HR+ , what is HR-?

We are grateful to the Referee for pointing out this issue. We have now added definition of HR+ and HR- terms to the revised version of the manuscript. (line 237 in the revised version of the manuscript). It now reads:

Further, individual lipid intensities correlated with HR well above the chance expectation (HR lipids, $N = 60$, $lm(\text{lipid intensity} \sim \text{hierarchy level})$, BH-corrected $p < 0.05$; Table S3), 26 of them positively (HR+ lipids) and 34 – negatively (HR– lipids).

In Results, the authors need to state upfront that the MRI Was from different human and macaque subjects (not the same ones used for the lipidomic analysis). This remains a major limitation of this study.

We appreciate the Referee's comment. In response to the Referee's suggestion, we have clarified in the Results section that the structural MRI data was recorded from a different set of human and macaque subjects than those used for lipidome analysis (line 144 in the revised version of the manuscript). It now reads:

For both species, myelin content distribution correlated strongly with the first principal component of the lipidome intensity distribution, even though the lipid and sMRI data were acquired from entirely distinct sets of individuals (Pearson's $R = 0.78$ and 0.77 for humans and macaques, respectively, $p < 0.0001$; Figure 3C,D).

We, however, do not feel that this fact limits the conclusions of our study. On the contrary, we argue that the observed significant and strong positive correlation between the variation pattern of lipids, known to be associated with myelin membranes in our mass spectrometry-based data derived from one set of brains, and variation in myelin intensity across brain regions detected using structural MRI methods in the other set of brains, suggests that lipid intensity differences among brain regions reported in our study represent a general phenomenon shared among individuals of a species. Otherwise, results could have been interpreted as limited to the specific set of brain samples used in a given study.

In the discussion, the authors state: "This category, termed housekeeping lipids, was 279 significantly enriched in short-chain unesterified fatty acids, aligning with their role as universal building blocks in brain lipid biosynthesis". This is simply not true (the review paper cited does not show that either). Also, which short chain unsaturated fatty acids are they referring to?

We are grateful to the Referee for pointing out this issue. As indicated by the Referee, our statistical analysis reveals a significant overrepresentation of all non-esterified (free) fatty acids, when considered as a class, in the "housekeeping" lipid category, not just short-chain ones. Within the lipid class, our evaluation of chain length and unsaturation demonstrates that housekeeping lipids as a category included unsaturated (Mann–Whitney U test, BH-corrected $p < 0.05$; Figure 4A) and short-chain free fatty acids (Mann–Whitney U test, BH-corrected $p < 0.1$; Figure 4A). We have clarified this point in the revised version of the manuscript (line 179 in the revised manuscript), which now reads:

By contrast, housekeeping lipids as a category included unsaturated (Mann–Whitney U test, BH-corrected $p < 0.05$; Figure 4A) and short-chain free fatty acids (FA) (Mann–Whitney U test, BH-corrected $p < 0.1$; Figure 4A).

REVIEWERS' COMMENTS

Reviewer #3 (Remarks to the Author):

Authors have addressed my concerns.

Editorial comments in the Reporting Summary:

Please see the author guidance document for a list of figures and figure legends that require additional information.

We have updated all figures and figure legends that required additional information. Please find point-by-point answers in the attached author checklist.

Please ensure all the data collection/data analysis software/tools/algorithms/packages used in the study are clearly mentioned in the manuscript and are also listed here in the reporting summary (with version numbers).

We have double-checked that all the data collection and data analysis tools used in the study are mentioned in the manuscript and in the reporting summary (with version numbers). No changes are needed.

Since custom codes have been developed in the study, we strongly recommend that the custom codes be deposited in a public DOI-minting repository such as Zenodo or Code Ocean and cited in the reference list. Authors are encouraged to manage subsequent code versions and to use a license approved by the open source initiative. Full details about how the code can be accessed and any restrictions must be described in the Code Availability statement of the manuscript and here in the RS as well.

We have deposited our custom code to Zenodo. Here is the DOI: [10.5281/zenodo.10908108](https://doi.org/10.5281/zenodo.10908108). We have added the following Code Availability statement to the manuscript and to the reporting summary: "All custom code used in this manuscript is publicly available at <https://doi.org/10.5281/zenodo.10908108>."

Please mention all the databases/datasets used in the study along with appropriately accessible links/accession codes in the manuscript under the "Data availability" section as well as in this reporting summary. For example: The Atlas of the Human Brain, etc.

We have added this information to the Data Availability section and reporting summary as follows: "The RNA-data data used in this study for the remaining 33 brain regions are available under accession number GSE127898 [<https://www.ncbi.nlm.nih.gov/geo/query/acc.cgi?acc=GSE127898>]."

Please elaborate on the covariate-relevant population characteristics of the human research participants (e.g. past and current diagnosis, etc.).

We have included the following additional information to the reporting summary: "All human subjects were cognitively healthy and did not have any brain-related diagnoses. All human subjects suffered sudden death with no prolonged agony state from causes not related to brain function." This information is already present in the manuscript, thus no changes are needed there.

Please provide the full names of the boards/committees that approved the protocol, in the reporting summary as well as in the manuscript.

We have included the full name of the committee that approved the protocol, in the reporting summary as well as in the manuscript, as follows: “The protocol was approved by the Skoltech Institutional Review Board.”

Please elaborate on how sample sizes were chosen (providing relevant citations if available).

We have elaborated on the sample size choice and provided relevant citations in the reporting summary: “We performed an exploratory study and the sample size was determined based on the published studies constructing the human brain transcriptome maps as the closest available analog (Khrameeva et al., 2020, <https://doi.org/10.1101/gr.256958.119>). We measured the lipidome composition in 75 distinct brain regions dissected from four humans and 38 brain three macaques, which totals to >400 samples, by two mass spectrometry-based techniques (HRMS and MRM). Because each measurement takes >30 min (30 min x 400 samples = 200 hours), excluding the time for the sample preparation and lipid extraction, analyzing a greater number of samples would not be feasible, and running measurement on different machines in parallel would create batch effects. Therefore, the samples sizes were chosen as maximum possible ones for the analysis withing reasonable timeframes.”

Please state how often the experiments were replicated or performed independently.

We have included the requested information to the reporting summary as follows: “The human lipidome data was produced from 4 biological replicates. We measured reproducibility by performing principal component analysis (PCA). PCA demonstrated a clear separation by brain regions rather than the replicates. In addition, the results obtained on this main dataset were replicated in a series of independent follow-up experiments. In particular, macaque brain samples were used to replicate the results obtained from human brain data as macaques were raised in a standardized environment, followed by rapid and controlled tissue collection, thus providing a controlled reference for the human brain lipidome quality evaluation. The macaque lipidome data was produced from 3 biological replicates. Additionally, two independent mass-spectrometry techniques (HRMS and MRM) were used to replicate both human and macaque results. We independently measured lipid composition by HRMS and MRM, in 4 human biological replicates and 3 macaque biological replicates.”

Please specify the age of laboratory animals used, here in the reporting summary.

We have specified the age in the reporting summary as follows: “Adult Tg(Thy1-COP4/EYFP)9Gfng (Thy1-ChR2-YFP) transgenic mice (4-5 months).”

Please provide information on housing conditions for the mice, describing the dark/light cycle, ambient temperature, and humidity in the manuscript.

We have included this information to the manuscript as follows: "Mice were housed in standard breeding cages at constant temperature (22 ± 1 °C) and relative humidity (50%), with a 12:12 h light:dark cycle."

Please state here that no wild animals were used in the study.

We have included this statement as suggested.

Please state here that no field collected samples were used in the study.

We have included this statement as suggested.

Please provide the full names of the boards/committees that approved the protocol, in the reporting summary as well as in the manuscript.

We have included the full name of the committee that approved the protocol, in the reporting summary as well as in the manuscript, as follows: "The protocol was approved by the Skoltech Institutional Review Board in accordance with the guidelines on the ethical use of animals. All possible efforts were made to minimize animal suffering, and to reduce the number of animals used per condition by calculating the necessary sample size before performing the experiments."